# T2T: From Distribution Learning in Training to Gradient Search in Testing for Combinatorial Optimization

**Yang Li, Jinpei Guo, Runzhong Wang, Junchi Yan**[*]
Dept. of Computer Science and Engineering & MoE Key Lab of AI
Shanghai Jiao Tong University
{yanglily, jinpei, runzhong.wang, yanjunchi}@sjtu.edu.cn

## Abstract

Extensive experiments have gradually revealed the potential performance bottleneck of modeling Combinatorial Optimization (CO) solving as neural solution prediction tasks. The neural networks, in their pursuit of minimizing the average objective score across the distribution of historical problem instances, diverge from the core target of CO of seeking optimal solutions for every test instance. This calls for an effective search on each problem instance, while the model should serve to provide supporting knowledge that benefits the search. To this end, we propose *T2T (Training to Testing)* framework that first leverages the generative modeling to estimate the high-quality solution distribution for each instance during training, and then conducts a gradient-based search within the solution space during testing. The proposed neural search paradigm consistently leverages generative modeling, specifically diffusion, for graduated solution improvement. It disrupts the local structure of the given solution by introducing noise and reconstructs a lower-cost solution guided by the optimization objective. Experimental results on Traveling Salesman Problem (TSP) and Maximal Independent Set (MIS) show the significant superiority of T2T, demonstrating an average performance gain of 49.15% for TSP solving and 17.27% for MIS solving compared to the previous state-of-the-art.

## 1 Introduction

Machine Learning (ML) for Combinatorial Optimization (CO)[2], abbreviated as ML4CO [1; 2; 3; 4], is a rapidly growing area, lying at the intersection between the two well-established communities. CO problems are essential in operational research as well as computer science, in both theory and practice. However, due to the inherent computational difficulty, e.g. NP-hardness, (approximately) solving these problems with effectiveness and efficiency poses significant challenges. Compared with traditional heuristic-based solvers, ML has recently shown its promising potential to automatically learn heuristics in a data-driven scheme [1; 2; 3; 4], or meanwhile with the aid of human knowledge which would normally improve the sample-efficiency [5; 6; 7; 8; 9]. Moreover, the introduction of neural networks also allows for the speedup by parallel matrix computation in forward inference.

A popular framework in ML4CO is developing neural networks to directly predict a solution or generate a solution with the aid of neural predictions, such that the corresponding objective score is minimized [10; 11; 12; 13]. However, the neural network can only fit limited mappings from problem

---

[*]Correspondence author. This work was partly supported by National Key Research and Development Program of China (2020AAA0107600), NSFC (62222607) and STCSM (22511105100).

[2]In this paper, CO problems mainly refer to optimization problems on graphs [1] with an objective function.

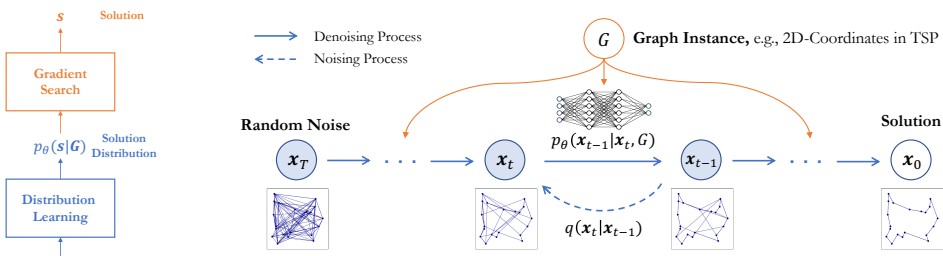

Figure 1: Diagram of proposed framework.

Figure 2: Diffusion modeling for CO solving where the model learns how to gradually denoise from the random noise to predict each step's data distribution, eventually modeling the solution distribution.

parameters to (near) optimal solutions in the training set, resulting in subpar performance on unseen test instances [14; 15]. Therefore, it calls for neural solvers to efficiently explore solutions tailored to each specific problem instance. In the traditional framework, the network prediction typically converges to a local optimum with little diversity, struggling to further explore superior solutions. In this paper, we resort to a more effective framework that first learns to model the distribution of high-quality solutions conditioned on problem instances, then conducts a gradient-based search to obtain a superior solution specific to the given instance, as shown in Fig. 1. Existing alternative methods that conduct the tailored search in the testing phase, e.g. active search [16; 17] and meta-learning [18], require updating the model weights during testing, hindering their applications in computational intensive scenarios. In contrast, we develop a more efficient gradient search framework to obtain a superior solution specific to the given instance, without the need to update model weights.

We exploit the powerful generative neural CO solver [19] that learns to model the distribution of high-quality solutions conditioned on specific problem instances. The generative models are welcomed in their ability to output high-quality images [20], texts [21] and graphs [22] with great diversity, whereby these advantages are also welcomed in the ML4CO community. Some recent successes in generative learning of CO are demonstrated in classic CO problems e.g. Traveling Salesman Problem (TSP) and Maximal Independent Set (MIS) (via Diffusion [19] and Variational Auto-Encoder [23]), Satisfiability Problem (SAT) and Mixed-Integer Linear Programming (MILP) (via graph generators [24; 25; 26]), also in more realistic applications e.g. routing for chip design (via Diffusion [27] and Generative Adversarial Network [28]). Notably, the very recent adaptation of the flourishing diffusion models in probabilistic generation [19] has resulted in state-of-the-art performance for solving CO problems e.g. TSP. The diffusion modeling for CO solving is illustrated in Fig. 2, where the model is designed to estimate the solution distribution given a certain instance $G$, i.e. $p_\theta(\mathbf{x}_0, G)$. Despite its high-quality solutions, a major drawback is the unawareness of optimizing the CO objective as the generative model simply learns to fit a distribution of solutions. To further improve the efficiency of the current costly from-scratch sampling, it stands crucial to introduce a more efficient search-based paradigm to further explore high-quality solutions in terms of the final objective of the problem.

To this end, we propose a gradient-based search paradigm to further leverage the learned solution distribution through training, bridging the solution distribution learning during generative modeling **t**raining **to** gradient search during **t**esting for **CO**, dubbed T2T. The proposed search paradigm well aligns with the behavior of a progressive generative model especially for the SOTA embodiment i.e. diffusion. The diffusion model provides a probabilistic framework that enables a smooth and flexible transition between random noises and feasible solutions, allowing for varying degrees of structure disruption, with a reference denoising process to restore a solution. Leveraging this framework, it is natural to model the local search heuristic as two stages, i.e., disrupting the local structure and reconstructing it with a better objective score. Specifically, T2T first adds certain noise to the given solution and then denoises it to reconstruct a lower-cost solution in iterations, which is achieved by incorporating the objective optimization guidance into the denoising steps through gradient feedback in the loop, steering the recovered sampling towards the optimization direction, as illustrated in Fig. 3. T2T enhances the solving paradigm by incorporating a search procedure with explicit objective guidance, which enables more efficient sampling and allows for deeper exploitation of the estimated solution distribution.

For the discrete nature of CO, T2T builds itself upon discrete diffusion models [29; 30; 31], which in training learns the transition between categorical noises and feasible discrete solutions that are

represented as probability maps of nodes/edges on graphs indicating whether each node/edge is included in the solution. The transition is tailored for each instance as it is conditioned on the corresponding instance's statistics. The training of T2T follows the standard training paradigm of diffusion models, which primarily equips the model with feasibility awareness and estimates the distribution of high-quality solutions. While during testing, we introduce the objective optimization guidance which serves as the complementary optimization and gradient search utility.

We showcase the efficacy of T2T on two typical NP-hard combinatorial optimization problems for edge-selecting and node-selecting types respectively, i.e., TSP and MIS, which have been attracting most attention in the community of ML4CO. Empirically, T2T exhibits strong performance superiority over existing state-of-the-art neural solvers on benchmark datasets across various scales. Notably, a significant performance gain is observed when compared to the purely constructive diffusion solver [19], thereby highlighting the effectiveness of the designed search-based paradigm.

The highlights of this paper include: 1) We introduce the framework of T2T which first estimates the high-quality solution distribution for each instance during training, and then leverages the diffusion process to perform an effective local search in the solution space during testing; 2) We incorporate the optimization objective guidance into the denoising model with theoretical analysis, which equips the model with gradient feedback directly from the instance-wise objective score, enabling it to perform optimization while ensuring general feasibility; 3) Extensive experiments show the promising performance of the proposed T2T, even when trained without high-quality solution labels.

## 2    Related Work

**Machine Learning for CO Solving.** ML solvers can be categorized into construction solvers and improvement solvers. Constructive solvers include autoregressive solvers [32; 12; 6; 23; 8] which step-by-step extend a partial solution to produce a complete feasible solution, and non-autoregressive solvers [33; 34; 14; 35; 19] which predict a solution (typically soft constrained) in one shot. Improvement solvers [36; 37; 5] often define local operators such as 2OPT [36; 37], node swapping [5; 37], sub-problem resolving [7; 38] to perform local search towards minimizing the optimization objective. The algorithm pipelines generally involve selecting specific regions and locally optimizing sub-solutions. While T2T can be a rather different improvement solver, which allows for parallel optimization across multiple sub-regions and equips the generative modeling with the search procedure.

Generative modeling for CO leverages the powerful expressive and modeling capabilities of generative models for direct CO solving (e.g. VAE [23; 39], GAN [28; 40; 41], Diffusion [19; 27; 42], as well as GFlowNets [43; 44], which, while not a typical generative model itself, is closely related to generative modeling), and for supporting CO solving [24; 25]. DIFUSCO [19] has become state-of-the-art for the TSP problem by leveraging powerful diffusion models. However, no instance-specific search paradigms are introduced in existing methods to fully utilize the estimated solution distribution. Instead, they rely on multiple from-scratch samplings to leverage the distribution, which is computationally expensive. Though [23] attempts to utilize the evolution algorithm to conduct multiple iterations of sampling, it yet incurs even more significant costs without explicit objective guidance in the loop. In contrast, T2T serves as an improvement solver that locally modifies the given solution with objective guidance in the loop. It offers the flexibility to introduce any degree of disruption and rewrite without the necessity to construct the solution from scratch.

**Diffusion Probabilistic Models.** Diffusion models (score-based models) typically comprise a noising process and a learnable denoising process, where neural networks predict the noise introduced in each step. Diffusion's inference can be seen as a score (gradient) guided search, which is a general principle across different settings [45; 46; 47]. Diffusion in continuous space (e.g., for image generation) [29; 48; 42; 49; 50; 51; 52] have established and consolidated the theoretical foundations of the general diffusion framework. Several landmark works have advanced the diffusion model through various aspects. DDIM [49] generalizes DDPM via a class of non-Markovian processes which gives rise to implicit models that produce high-quality samples much faster. For conditional generation, classifier guidance [29; 52] and classifier-free guidance [53] are introduced with the score estimate of the diffusion models to produce conditional results. Similar paradigms are adopted in discrete diffusion models for generating discrete data using binomial noises [29] or multinomial/categorial noises [30; 31]. Diffusion also demonstrates its capability to learn offline policies in reinforcement learning tasks [54; 55].

## 3 Approach

This section presents our proposed T2T framework, including the preliminaries, offline training pipeline based on generative modeling, and online testing pipeline with objective guided search.

### 3.1 Preliminaries: Combinatorial Optimization on Graphs

Following the notations adopted in [13; 56; 57] we define $\mathcal{G}$ as the universe of CO problem instances represented by graphs $G(V, E) \in \mathcal{G}$, where $V$ and $E$ denote the node set and edge set respectively. CO problems can be broadly classified into two types based on the solution composition: edge-selecting problems which involve selecting a subset of edges, and node-selecting problems which select a subset of nodes. Let $\boldsymbol{x} \in \{0, 1\}^N$ denote the optimization variable. For edge-selecting problems, $N = n^2$ and $\boldsymbol{x}_{i \cdot n + j}$ indicates whether $E_{ij}$ is included in $\boldsymbol{x}$. For node-selecting problems, $N = n$ and $\boldsymbol{x}_i$ indicates whether $V_i$ is included in $\boldsymbol{x}$. The feasible set $\Omega$ consists of $\boldsymbol{x}$ satisfying specific constraints as feasible solutions. A CO problem on $G$ aims to find a feasible $\boldsymbol{x}$ that minimize the given objective function $f(\cdot; G) : \{0, 1\}^N \to \mathbb{R}_{\geq 0}$:

$$\min_{\boldsymbol{x} \in \{0,1\}^N} f(\boldsymbol{x}; G) \quad \text{s.t.} \quad \boldsymbol{x} \in \Omega \tag{1}$$

In this paper, we study two primary and representative CO problems: TSP and MIS. TSP defines on an undirected complete graph $G = (V, E)$, where $V$ represents $n$ cities and each edge $E_{ij}$ has a non-negative weight $w_{ij}$ representing the distance between cities $i$ and $j$. The problem can then be formulated as finding a Hamiltonian cycle of minimum weight in $G$. For MIS, given an undirected graph $G = (V, E)$, an independent set is a subset of vertices $S \subseteq V$ such that no two vertices in $S$ are adjacent in $G$. MIS is the problem of finding an independent set of maximum cardinality in $G$.

### 3.2 Offline Solution Distribution Learning on Training Set based on Diffusion

The basic diffusion modeling for solution generation in training is primarily based on [19]'s design while this paper explicitly formulate the solving task as a conditional generation task. As mentioned in Sec. 3.1, the solutions of CO problems can be represented as $\boldsymbol{x} \in \{0, 1\}^N$ with $\boldsymbol{x} \in \Omega$. For each entry, the model estimates a Bernoulli distribution indicating whether this entry should be selected. In implementation, each entry of solution $\mathbf{x}$ is represented by a one-hot vector[3] such that $\mathbf{x} \in \{0, 1\}^{N \times 2}$. Generative modeling aims to model the distribution of high-quality solutions conditioned on the given instance, while feasibility constraints can be broadly captured through learning and eventually hard-guaranteed by post-processing. For the purpose of problem-solving, the generative model is desired to estimate the solution distribution given a certain instance $G$, i.e., $p_\theta(\mathbf{x}|G)$. In this section, we model this distribution through the discrete diffusion models [42; 30].

Following the notations of [42; 30], the general framework of diffusion includes a forward noising and a reverse denoising Markov process. The noising process, also denoted as the diffusion process, takes the initial solution $\mathbf{x}_0$ sampled from the distribution $q(\mathbf{x}_0|G)$ and progressively introduces noise to generate a sequence of latent variables $\mathbf{x}_{1:T} = \mathbf{x}_1, \mathbf{x}_2, \cdots, \mathbf{x}_T$. Specifically, the noising process is formulated as $q(\mathbf{x}_{1:T}|\mathbf{x}_0) = \prod_{t=1}^T q(\mathbf{x}_t|\mathbf{x}_{t-1})$. The denoising process is learned by the model, which starts from the final latent variable $\mathbf{x}_T$ and denoises $\mathbf{x}_t$ at each time step to generate the preceding variables $\mathbf{x}_{t-1}$ based on the instance $G$, eventually recovering the target data distribution. The formulation of the denoising process is expressed as $p_\theta(\mathbf{x}_{0:T}|G) = p(\mathbf{x}_T) \prod_{t=1}^T p_\theta(\mathbf{x}_{t-1}|\mathbf{x}_t, G)$, where $\theta$ denotes the parameters of the model. The training optimization aims to align $p_\theta(\mathbf{x}_0|G)$ with the data distribution $q(\mathbf{x}_0|G)$ and the objective adopts the variational upper bound of the negative log-likelihood:

$$\mathcal{L} = \mathbb{E}_q \left[ -\log p_\theta(\mathbf{x}_0|\mathbf{x}_1, G) + \sum_{t>1} D_{KL} \left[ q(\mathbf{x}_{t-1}|\mathbf{x}_t, \mathbf{x}_0) \parallel p_\theta(\mathbf{x}_{t-1}|\mathbf{x}_t, G) \right] \right] + C \tag{2}$$

Specifically, the forward noising process is achieved by multiplying $\mathbf{x}_t \in [0, 1]^{N \times 2}$ at step $t$ with a forward transition probability matrix $\mathbf{Q}_t \in [0, 1]^{2 \times 2}$ where $[\mathbf{Q}_t]_{ij}$ indicates the probability of transforming $\mathbf{e}_i$ in each entry of $\mathbf{x}_t$ to $\mathbf{e}_j$. We set $\mathbf{Q}_t = \begin{bmatrix} \beta_t & 1 - \beta_t \\ 1 - \beta_t & \beta_t \end{bmatrix}$ [30], where $\beta_t \in [0, 1]$

---

[3]Each entry with $[0, 1]$ indicates that it is included in $\mathbf{x}$ and $[1, 0]$ indicates the opposite.

such that the transition matrix is doubly stochastic with strictly positive entries, ensuring that the stationary distribution is uniform which is an unbiased prior for sampling. The noising process for each step and the $t$-step marginal are formulated as:

$$q(\mathbf{x}_t|\mathbf{x}_{t-1}) = \text{Cat}(\mathbf{x}_t; \mathbf{p} = \mathbf{x}_{t-1}\mathbf{Q}_t) \quad \text{and} \quad q(\mathbf{x}_t|\mathbf{x}_0) = \text{Cat}(\mathbf{x}_t; \mathbf{p} = \mathbf{x}_0\overline{\mathbf{Q}}_t) \tag{3}$$

where $\text{Cat}(\mathbf{x}; \mathbf{p})$ is a categorical distribution over $N$ one-hot variables with probabilities given by vector $\mathbf{p}$ and $\overline{\mathbf{Q}}_t = \mathbf{Q}_1\mathbf{Q}_2\cdots\mathbf{Q}_t$. Through Bayes' theorem, the posterior can be achieved as:

$$q(\mathbf{x}_{t-1}|\mathbf{x}_t, \mathbf{x}_0) = \frac{q(\mathbf{x}_t|\mathbf{x}_{t-1}, \mathbf{x}_0)q(\mathbf{x}_{t-1}|\mathbf{x}_0)}{q(\mathbf{x}_t|\mathbf{x}_0)} = \text{Cat}\left(\mathbf{x}_{t-1}; \mathbf{p} = \frac{\mathbf{x}_t\mathbf{Q}_t^\top \odot \mathbf{x}_0\overline{\mathbf{Q}}_{t-1}}{\mathbf{x}_0\overline{\mathbf{Q}}_t\mathbf{x}_t^\top}\right) \tag{4}$$

The neural network is trained to predict the logits of the distribution $\tilde{p}_\theta(\tilde{\mathbf{x}}_0|\mathbf{x}_t, G)$, such that the denoising process can be parameterized through $q(\mathbf{x}_{t-1}|\mathbf{x}_t, \tilde{\mathbf{x}}_0)$:

$$p_\theta(\mathbf{x}_{t-1}|\mathbf{x}_t) \propto \sum_{\tilde{\mathbf{x}}_0} q(\mathbf{x}_{t-1}|\mathbf{x}_t, \tilde{\mathbf{x}}_0)\tilde{p}_\theta(\tilde{\mathbf{x}}_0|\mathbf{x}_t, G) \tag{5}$$

Specifically for implementation, the condition $G$ is enforced as an input to the network $\theta$, which is embodied as an anisotropic graph neural network with edge gating mechanisms [33] following [19]. For TSP, the instance condition consists of 2D coordinates of the vertices. The network input includes node embeddings from the 2D coordinates, along with edge embeddings from $\mathbf{x}_t$. For MIS, the edges $E$ serve as the instance condition, complemented by the inclusion of node embeddings from $\mathbf{x}_t$ to collectively form the input. The output of the network is $\tilde{p}_\theta(\tilde{\mathbf{x}}_0|\mathbf{x}_t, G) \in [0, 1]^{N\times 2}$ parameterizing $N$ Bernoulli distributions for $N$ entries in $\tilde{\mathbf{x}}_0$.

### 3.3  Test-stage Gradient-based Search with Instance-wise Objective Feedback

In this section, we first present the technical solution for the objective gradient incorporation in denoising steps, and then illustrate the neural search paradigm that leverages the guided denoising steps.

#### 3.3.1  Objective-aware Denoising Process with Gradient Feedback

In the context of combinatorial optimization, the incorporation of objective optimization is necessary and important, which enables the direct involvement of objective and effective search over the solution space towards minimizing the score. As discussed in Sec. 3.2, at step $t$ the model denoise $\mathbf{x}_t$ by $p_\theta(\mathbf{x}_{t-1}|\mathbf{x}_t, G)$. For the purpose of objective optimization, we aim to estimate $p_\theta(\mathbf{x}_{t-1}|\mathbf{x}_t, G, y^*)$ where $y^*$ is the optimal objective score given the instance $G$:

$$y^* = \min_{\mathbf{x}} l(\mathbf{x}; G) \quad \text{s.t.} \quad \mathbf{x} \in \Omega \tag{6}$$

to guide the denoising process to $\mathbf{x}^* = \arg\min_{\mathbf{x}} l(\mathbf{x}; G)$. We will show later that this ability does not necessarily require the access of $y^*$ or $\mathbf{x}^*$, and can readily be achieved by the model learned in Sec. 3.2 without relying on any additionally trained networks. The adopted objective is defined below.

**Optimization Objective.**  Since the generated samples are not guaranteed to satisfy the hard constraints, we incorporate constraint satisfaction into the objective and utilize a relaxed objective with constraint penalty, as adopted in [13; 56; 57]. Consider a relaxed cost function $f_r(\cdot; G) : \{0, 1\}^N \to \mathbb{R}_{\geq 0}$ which defined on the constraint relaxed space $\{0, 1\}^N$ satisfying $f_r(\boldsymbol{x}; G) = f(\boldsymbol{x}; G), \forall \boldsymbol{x} \in \Omega$, and a constraint penalty $g_r(\boldsymbol{x}; G) : \{0, 1\}^N \to \mathbb{R}_{\geq 0}$ where $g_r(\boldsymbol{x}; G) = 0$ if $\boldsymbol{x} \in \Omega$ and $g_r(\boldsymbol{x}; G) \geq 1$ if $\boldsymbol{x} \notin \Omega$, the objective can be formulated as:

$$l(\boldsymbol{x}; G) \triangleq f_r(\boldsymbol{x}; G) + \beta g_r(\boldsymbol{x}; G) \tag{7}$$

where $\beta > \max_{\boldsymbol{x} \in \Omega} f(\boldsymbol{x}; G)$. For MIS, the constraint can be straightforwardly modeled as $\boldsymbol{x}_i\boldsymbol{x}_j = 0$ if $(i, j) \in E$, thus the objective can be formulated as $l_{\text{MIS}}(\boldsymbol{x}; G) \triangleq -\sum_{1\leq i\leq N}\boldsymbol{x}_i + \beta\sum_{(i,j)\in E}\boldsymbol{x}_i\boldsymbol{x}_j$. While for TSP, since the constraints are difficult to explicitly express, we directly select the objective as $l_{\text{TSP}} = \boldsymbol{x} \odot D$ where $D \in \mathbb{R}_+^{n\times n}$ denotes the distance matrix.

**Objective Guidance in Denoising Steps.**  The estimation of $p_\theta(\mathbf{x}_{t-1}|\mathbf{x}_t, G, y^*)$ can be modeled through the following proposition, adapted from the classifier guidance technique [52]:

**Proposition 1.** *The optimization-enforced denoising probability estimation $p_\theta(\mathbf{x}_{t-1}|\mathbf{x}_t, G, y^*)$ equals to $Zp_\theta(\mathbf{x}_{t-1}|\mathbf{x}_t, G)p(y^*|\mathbf{x}_{t-1}, G)$, where $Z$ is a normalizing constant.*

The proof is given in the Appendix. As $p_\theta(\mathbf{x}_{t-1}|\mathbf{x}_t, G)$ can be readily obtained from the trained network, the main challenge of estimating $p_\theta(\mathbf{x}_{t-1}|\mathbf{x}_t, G, y^*)$ lies in estimating $p(y^*|\mathbf{x}_{t-1}, G)$. Since $\mathbf{x}_{t-1}$ is not accessible at step $t$, we utilize Taylor expansion to approximate $\log p(y^*|\mathbf{x}_{t-1}, G)$ around $\mathbf{x}_{t-1} = \mathbf{x}_t$, given that $\mathbf{x}_{t-1} \sim \mathbf{x}_t$:

$$\log p(y^*|\mathbf{x}_{t-1}, G) \approx \log p(y^*|\mathbf{x}_t, G) + \left[\nabla_{\mathbf{x}_t} \log p(y^*|\mathbf{x}_t, G)\right]^\top (\mathbf{x}_{t-1} - \mathbf{x}_t)$$

$$= \left[\nabla_{\mathbf{x}_t} \log p(y^*|\mathbf{x}_t, G)\right]^\top \mathbf{x}_{t-1} + \underbrace{\log p(y^*|\mathbf{x}_t, G) - \left[\nabla_{\mathbf{x}_t} \log p(y^*|\mathbf{x}_t, G)\right]^\top \mathbf{x}_t}_{C(\mathbf{x}_t)} \quad (8)$$

where $C(\mathbf{x}_t)$ is irrelevant to $\mathbf{x}_{t-1}$. Applying exponentiation, we obtain:

$$p(y^*|\mathbf{x}_{t-1}, G) \propto \exp\left(\left[\nabla_{\mathbf{x}_t} \log p(y^*|\mathbf{x}_t, G)\right]^\top \mathbf{x}_{t-1}\right) \quad (9)$$

To determine $p(y^*|\mathbf{x}_t, G)$, we utilize energy-based modeling [58] to model the distribution $p(y|\mathbf{x}_t, G)$. In this context, we define the energy function as:

$$E(y, \mathbf{x}_t, G) = |y - l(\mathbf{x}_0(\mathbf{x}_t); G)| \quad (10)$$

where $\mathbf{x}_0(\mathbf{x}_t)$ denotes the clean target sample $\mathbf{x}_0$ corresponding to $\mathbf{x}_t$. The energy function quantifies the compatibility between $y$ and $(\mathbf{x}_t, G)$, and it reaches zero when $y$ is exactly the final objective score originating from $\mathbf{x}_t$ with respect to $G$. Such a design enables the best $y$ matching the inputs to maintain the highest probability density, and the probability density is positively correlated with the matching degree. Then we employ the *Gibbs distribution* to characterize the probability distribution over a collection of arbitrary energies:

$$p(y|\mathbf{x}_t, G) = \frac{\exp(-E(y, \mathbf{x}_t, G))}{\int_{y'} \exp(-E(y', \mathbf{x}_t, G))} = \frac{\exp(-|y - l(\mathbf{x}_0(\mathbf{x}_t); G)|)}{\int_{y'} \exp(-|y' - l(\mathbf{x}_0(\mathbf{x}_t); G)|)} \quad (11)$$

Let $Z = \int_{y'} \exp(-|y' - l(\mathbf{x}_0(\mathbf{x}_t); G)|)$ and substitute in $y^*$, we have:

$$\log p(y^*|\mathbf{x}_t, G) = y^* - l(\mathbf{x}_0(\mathbf{x}_t); G) - \log Z \quad \text{and} \quad \nabla_{\mathbf{x}_t} \log p(y^*|\mathbf{x}_t, G) = -\nabla_{\mathbf{x}_t} l(\mathbf{x}_0(\mathbf{x}_t); G) \quad (12)$$

Note the gradient operator has removed to effect of $y^*$ as a constant, thereby eliminating the need to access $y^*$ for achieving the target. And $Z$ as the integral over the entire distribution is not affected by $\mathbf{x}_t$ and thus can also be removed by the gradient operator. Recall that the network $\theta$ is trained to predict the logits of $\tilde{p}_\theta(\tilde{\mathbf{x}}_0|\mathbf{x}_t) \in [0, 1]^{N \times 2}$ which includes the logits of $N$ Bernoulli distributions, satisfying $[\tilde{p}_\theta(\tilde{\mathbf{x}}_0|\mathbf{x}_t)]_i = \left(1 - [\mathbb{E}\tilde{\mathbf{x}}_0]_i, [\mathbb{E}\tilde{\mathbf{x}}_0]_i\right)$. Thus $l(\mathbf{x}_0(\mathbf{x}_t); G)$ can be estimated using $\mathbb{E}\tilde{\mathbf{x}}_0$ as:

$$l(\mathbf{x}_0(\mathbf{x}_t); G) \approx l(\mathbb{E}_{\tilde{\mathbf{x}}_0 \sim \tilde{p}_\theta(\tilde{\mathbf{x}}_0|\mathbf{x}_t)} \tilde{\mathbf{x}}_0; G) \quad (13)$$

By Eq. 9, 12 and 13, we obtain $p(y^*|\mathbf{x}_{t-1}, G) \propto \exp\left(\left[-\nabla_{\mathbf{x}_t} l(\mathbb{E}_{\tilde{\mathbf{x}}_0 \sim \tilde{p}_\theta(\tilde{\mathbf{x}}_0|\mathbf{x}_t)} \tilde{\mathbf{x}}_0; G)\right]^\top \mathbf{x}_{t-1}\right)$, and the guided denoising probability estimation can now be achieved by $p_\theta(\mathbf{x}_{t-1}|\mathbf{x}_t, G, y^*) = Z p_\theta(\mathbf{x}_{t-1}|\mathbf{x}_t, G) p(y^*|\mathbf{x}_{t-1}, G)$ where $Z$ is the normalizing constant, to realize the guided denoising.

### 3.3.2 The T2T Framework for Efficient Differentiable Neural Search

With the objective gradient incorporated denoising process in Sec. 3.3.1, we now delve into T2T.

**Overview.** As shown in Fig. 3, the algorithm starts with an initial solution $\mathbf{s}_0$ and conducts several iterations to enhance the given solution. Each iteration involves adding a certain degree of noise to disrupt the structure, denoising with objective gradient guidance to obtain a lower-cost soft-constrained solution, and subsequently post-processing to decode a feasible solution $\mathbf{s} \in \Omega$. The algorithm eventually reports the solution with the lowest objective score ever achieved.

**Structure Disruption.** Recall that $\mathbf{x}_1, \mathbf{x}_2, \cdots, \mathbf{x}_T$ are situated in spaces characterized by progressively escalating levels of disruption. To introduce a controlled degree of disruption to the given solution, we employ $\mathbf{s}_0 \overline{\mathbf{Q}}_{\alpha T}$ to derive the distribution of the disrupted solution $q(\mathbf{s}_{\alpha T}|\mathbf{s}_0)$ as $N$ Bernoulli distributions, where $\alpha$ serves as a hyperparameter to control the degree of noise. Subsequently, the disrupted solution $\mathbf{s}_{\alpha T}$ can be sampled from $q(\mathbf{s}_{\alpha T}|\mathbf{s}_0)$.

**Objective Guided Solution Reconstruction.** From $\mathbf{s}_{\alpha T}$, we employ $p_\theta(\mathbf{s}_{t-1}|\mathbf{s}_t, G, y^*)$ proposed in Sec. 3.3.1 to perform denoising, which aimed at recovering a potentially lower-cost $\mathbf{s}_0'$. Additionally,

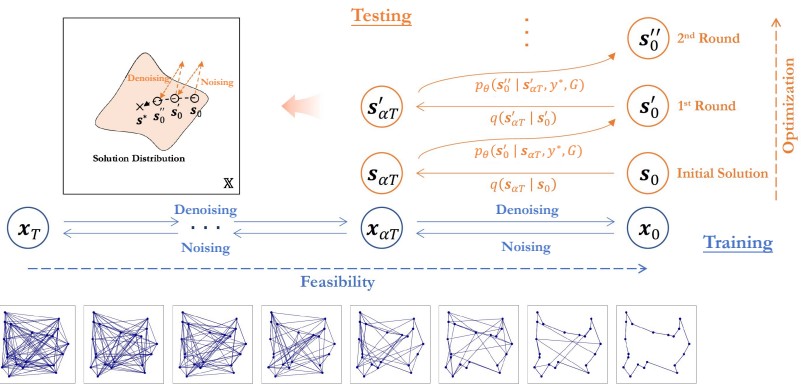

Figure 3: T2T iteratively adds noise to the solution and performs denoising guided by the problem objective (e.g. TSP), to improve the solution as shown in the orange part. The bottom blue part shows the capability of diffusion to establish a smooth transition from random noises to feasible solutions. $\mathbf{s}^*$: optimal solution; $y^*$: optimal objective score; $\theta$: model parameters; $G$: given instance.

we utilize the fast sampling algorithm introduced in DDIM [49] to accelerate sampling. The algorithm determines a $M$-element subset from latent variables $\mathbf{s}_{1:\alpha T}$ to obtain $\{\mathbf{s}_{\tau_1}, \mathbf{s}_{\tau_2}, \cdots, \mathbf{s}_{\tau_M}\}$ ($\tau_1 = 1$ and $\tau_M = \alpha T$), and directly models $p_\theta(\mathbf{s}_{\tau_{i-1}}|\mathbf{s}_{\tau_i}, G, y^*)$, which effectively reduces the number of denoising steps. In our implementation, we uniformly perform a 10-step reconstruction, i.e., $M = 10$.

**Post-inference Decoding.** Note the mechanism guarantees that the recovered solution $\mathbf{s} \in \{0,1\}^{N \times 2}$ (consisting of $N$ one-hot vectors), but it does not necessarily satisfy $\mathbf{s} \in \Omega$. Thus, in the last step's inference, we directly output the logits of its distribution $p_\theta(\mathbf{s}'_0)$ to produce $\mathbb{E}\mathbf{s}'_0$ as the heatmap $H \in [0,1]^N$ where each element denotes each edge/node's confidence to be selected, and then we adopt greedy decoding to obtain a feasible solution. We follow previous works [59; 35; 19] to perform greedy decoding by sequentially inserting edges (for TSP) or nodes (for MIS) with the highest confidence if there are no conflicts. For TSP, the 2OPT heuristic [60] is optionally applied.

**Tradeoff Between Guidance Accuracy and Local Rewriting Degree.** Note that the objective-guided denoising process relies heavily on the accuracy of the estimation $\tilde{p}_\theta(\tilde{\mathbf{x}}_0|\mathbf{x}_t)$. Intuitively, it is easier to restore the original solution from the data that has lower noise content. And with excessive noise interference, the estimation will deviate significantly from the ground truth, resulting in faulty guidance. This is also one of the reasons why we use a local rewriting mechanism. However, while the accuracy of the estimation increases as it moves closer to the original solution space, it also limits the degree of reconstruction in terms of the solution structure. Therefore, it is crucial to strike a balance between the estimation accuracy and the extent of local rewriting. To accomplish this, we employ the aforementioned hyperparameter $\alpha$ to control an appropriate range of perturbation.

## 4 Experiments

All the experiments are performed on GPUs of NVIDIA Telsla A100. All the test evaluations are performed on a single GPU, while the trainings are conducted in parallel with four A100 GPUs. Source code is publicly available at `https://github.com/Thinklab-SJTU/T2TCO`.

### 4.1 Experimental Setup

We test on two CO problems including TSP and MIS. The comparison includes SOTA learning-based solvers, heuristics, and exact solvers for each problem. For comparison fairness of learning solvers, the initial solutions of our approach are obtained by denoising random noises by Eq. 5 (same solution as the constructive diffusion solver DIFUSCO [19]). To balance the inference time, for experiments with runtime measurement, T2T utilizes a reduced number of sampling steps for constructing the initial solutions, specifically 20 steps. While DIFUSCO, as an important baseline, adopts 120 inference steps for TSP and 50 steps for MIS. The models for both TSP and MIS are trained with 1000 denoising steps, i.e., $T = 1000$. For gradient search, T2T generally involves 3 iterations with 10 guided denoising steps for each. See more details in Appendix.

Table 1: ***Greedy Decoding*** on TSP-50 and TSP-100. RL: Reinforcement Learning, SL: Supervised Learning, Grdy: Greedy Decoding. * denotes the baseline for the performance drop.

| ALGORITHM | TYPE | TSP-50 | | TSP-100 | |
|---|---|---|---|---|---|
| | | LENGTH↓ | DROP↓ | LENGTH↓ | DROP↓ |
| Concorde [62] | Exact | 5.69* | 0.00% | 7.76* | 0.00% |
| 2OPT [64] | Heuristics | 5.86 | 2.95% | 8.03 | 3.54% |
| Farthest Insertion | Heuristics | 6.12 | 7.50% | 8.72 | 12.36% |
| AM [12] | RL+Grdy | 5.80 | 1.76% | 8.12 | 4.53% |
| GCN [33] | SL+Grdy | 5.87 | 3.10% | 8.41 | 8.38% |
| Transformer [65] | RL+Grdy | 5.71 | 0.31% | 7.88 | 1.42% |
| POMO [6] | RL+Grdy | 5.73 | 0.64% | 7.84 | 1.07% |
| Sym-NCO [8] | RL+Grdy | – | – | 7.84 | 0.94% |
| Image Diffusion [59] | SL+Grdy | 5.76 | 1.23% | 7.92 | 2.11% |
| DIFUSCO [19] | SL+Grdy | 5.72 | 0.48% | 7.84 | 1.01% |
| T2T (Ours) | SL+Grdy | **5.69** | **0.04%** | **7.77** | **0.18%** |
| AM [12] | RL+Grdy+2OPT | 5.77 | 1.41% | 8.02 | 3.32% |
| GCN [33] | SL+Grdy+2OPT | 5.70 | 0.12% | 7.81 | 0.62% |
| Transformer [65] | RL+Grdy+2OPT | 5.70 | 0.16% | 7.85 | 1.19% |
| POMO [6] | RL+Grdy+2OPT | 5.73 | 0.63% | 7.82 | 0.82% |
| Sym-NCO [8] | RL+Grdy+2OPT | – | – | 7.82 | 0.76% |
| DIFUSCO [19] | SL+Grdy+2OPT | 5.69 | 0.09% | 7.78 | 0.22% |
| T2T (Ours) | SL+Grdy+2OPT | **5.69** | **0.02%** | **7.76** | **0.06%** |

## 4.2 Experiments for TSP

**Datasets.** A TSP instance includes $N$ 2-D coordinates and a reference solution obtained by heuristics. Training and testing instances are generated via uniformly sampling $N$ nodes from the unit square $[0, 1]^2$, which is a standard procedure as adopted in [12; 23; 33; 61; 35; 19]. We experiment on various problem scales including TSP-50, TSP-100, TSP-500, and TSP-1000. The reference solutions for TSP-50/100 are labeled by the Concorde exact solver [62] and the solutions for TSP-500/1000 are labeled by the LKH-3 heuristic solver [63]. The test set for TSP-50/100 is taken from [12; 33] with 1280 instances and the test set for TSP-500/1000 is from [34] with 128 instances for the fair comparison. This section also incorporates results on real-world TSPLIB[4] dataset.

**Metrics.** Following [12; 33; 35; 19], we adopt three evaluation metrics: 1) Length: the average total distance or cost of the solved tours w.r.t. the corresponding instances, as directly corresponds to the objective. 2) Drop: the relative performance drop w.r.t. length compared to the global optimality or the reference solution; 3) Time: the average computational time to solve the problems.

**Results for TSP-50/100.** Table 1 presents the results. Since DIFUSCO [19] has been shown able to attain global optimality for instances with these scales, at the cost of a sampling decoding scheme, we degrade to the greedy decoding setting for a more meaningful evaluation. The baselines include state-of-the-art learning-based methods with greedy decoding, as well as traditional exact and heuristic solvers. Hyperparameter $\alpha$ is set as $0.25$. Both DIFUSCO and T2T adopt 50 inference steps. As evident from the results, in the setting of pure greedy decoding without 2OPT refinement, the proposed search-based paradigm achieves a substantial reduction in performance drops from 0.48% to 0.04% for TSP-50 and from 1.01% to 0.18% for TSP-100.

**Results for TSP-500/1000.** Table 2 presents the results. The baselines include recent learning methods with greedy decoding and sampling decoding ($\times 4$), i.e. producing multiple solutions in parallel and selecting the best one, as well as traditional exact and heuristic solvers. DIFUSCO and T2T are compared in the same conditions. DIFUSCO and T2T adopt 120 and 20 inference steps, respectively. The results of other baselines are quoted from [34] and [35], with the runtimes provided for reference. In the sampling decoding setting, compared to previous neural solvers, T2T improves the performance drop from 0.83% to 0.37% for TSP-500 and 1.30% to 0.78% for TSP-1000.

**Effect of Disruption Ratio $\alpha$.** As discussed in Sec. 3.3.2, the hyperparameter $\alpha$ is utilized to regulate the extent of perturbation, striking a balance between guidance accuracy and local rewriting degree. Fig. 4 (a) illustrates the impact of $\alpha$ on the solving performance. The experiments are performed on TSP-500 with both greedy and sampling decoding settings. The results demonstrate the tradeoff existence and indicate that $0.3 \leq \alpha \leq 0.4$ can attain relatively better performance.

---

[4]http://comopt.ifi.uni-heidelberg.de/software/TSPLIB95/

Table 2: Results on TSP-500 and TSP-1000. AS: Active Search, S: Sampling Decoding, BS: Beam Search. * denotes the baseline for computing the performance drop. DIFUSCO and T2T are compared in the same running settings, while other numbers are quoted from [34; 35].

| Algorithm | Type | TSP-500 | | | TSP-1000 | | |
|---|---|---|---|---|---|---|---|
| | | Length↓ | Drop↓ | Time | Length↓ | Drop↓ | Time |
| Concorde [62] | Exact | 16.55* | – | 37.66m | 23.12* | – | 6.65h |
| Gurobi [66] | Exact | 16.55 | 0.00% | 45.63h | – | – | – |
| LKH-3 (default) [63] | Heuristics | 16.55 | 0.00% | 46.28m | 23.12 | 0.00% | 2.57h |
| Farthest Insertion | Heuristics | 18.30 | 10.57% | 0s | 25.72 | 11.25% | 0s |
| AM [12] | RL+Grdy | 20.02 | 20.99% | 1.51m | 31.15 | 34.75% | 3.18m |
| GCN [33] | SL+Grdy | 29.72 | 79.61% | 6.67m | 48.62 | 110.29% | 28.52m |
| POMO+EAS-Emb [17] | RL+AS+Grdy | 19.24 | 16.25% | 12.80h | – | – | – |
| POMO+EAS-Tab [17] | RL+AS+Grdy | 24.54 | 48.22% | 11.61h | 49.56 | 114.36% | 63.45h |
| DIMES [35] | RL+Grdy | 18.93 | 14.38% | 0.97m | 26.58 | 14.97% | 2.08m |
| DIMES [35] | RL+AS+Grdy | 17.81 | 7.61% | 2.10h | 24.91 | 7.74% | 4.49h |
| DIFUSCO [19] | SL+Grdy | 18.11 | 9.41% | 5.70m | 25.72 | 11.24% | 17.33m |
| T2T (Ours) | SL+Grdy | 17.39 | 5.09% | 4.90m | 25.17 | 8.87% | 15.66m |
| DIMES [35] | RL+Grdy+2OPT | 17.65 | 6.62% | 1.01m | 24.83 | 7.38% | 2.29m |
| DIMES [35] | RL+AS+Grdy+2OPT | 17.31 | 4.57% | 2.10h | 24.33 | 5.22% | 4.49h |
| DIFUSCO [19] | SL+G+2OPT | 16.81 | 1.55% | 5.75m | 23.55 | 1.86% | 17.52m |
| T2T (Ours) | SL+G+2OPT | **16.68** | **0.78%** | 4.98m | **23.41** | **1.25%** | 15.90m |
| EAN [67] | RL+S+2OPT | 23.75 | 43.57% | 57.76m | 47.73 | 106.46% | 5.39h |
| AM [12] | RL+BS | 19.53 | 18.03% | 21.99m | 29.90 | 29.23% | 1.64h |
| GCN [33] | SL+BS | 30.37 | 83.55% | 38.02m | 51.26 | 121.73% | 51.67m |
| DIMES [35] | RL+S | 18.84 | 13.84% | 1.06m | 26.36 | 14.01% | 2.38m |
| DIMES [35] | RL+AS+S | 17.80 | 7.55% | 2.11h | 24.89 | 7.70% | 4.53h |
| DIFUSCO [19] | SL+S | 17.48 | 5.65% | 19.02m | 25.11 | 8.61% | 59.18m |
| T2T (Ours) | SL+S | **17.02** | **2.84%** | 15.98m | **24.72** | **6.92%** | 53.92m |
| DIMES [35] | RL+S+2OPT | 17.64 | 6.56% | 1.10m | 24.81 | 7.29% | 2.86m |
| DIMES [35] | RL+AS+S+2OPT | 17.29 | 4.48% | 2.11h | 24.32 | 5.17% | 4.53h |
| DIFUSCO [19] | SL+S+2OPT | 16.69 | 0.83% | 19.05m | 23.42 | 1.30% | 59.53m |
| T2T (Ours) | SL+S+2OPT | **16.61** | **0.37%** | 16.03m | **23.30** | **0.78%** | 54.67m |

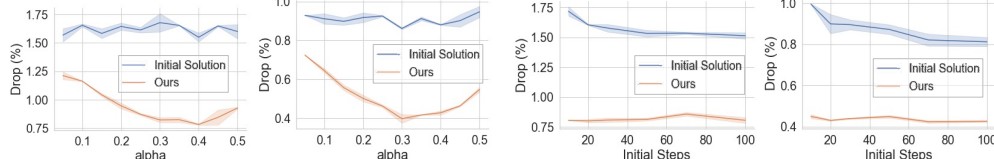

(a) $\alpha$-Drop Curve (Greedy and Sampling)    (b) InitStep-Drop Curve (Greedy and Sampling)

Figure 4: Effect of $\alpha$ and initialization steps to the performance drop. For each subgraph, the left represents the greedy decoding, while the right represents the sampling decoding.

**Effect of Initial Solution Quality.** Fig. 4 (b) illustrates the performance variation when altering the initial solution quality. The quality is controlled by the number of denoising steps for producing the initial solution. The experiments are performed on TSP-500 with both greedy and sampling decoding settings. As shown, T2T exhibits a relatively stable performance across varying initial solution qualities, based on which we generally utilize 20 steps for constructing the initial solutions.

**Generalization Results on Synthetic Data and TSPLIB.** Based on the problem set {TSP-50, TSP-100, TSP-500, TSP-1000}, we train the model on a specific problem scale and then evaluate it on all problem scales. Table 4 presents the generalization results of T2T and DIFUSCO with greedy decoding. The results show the satisfying cross-domain generalization ability of T2T, e.g., the model trained on TSP-1000 achieves less than a 2% optimality gap on all other problem scales. Moreover, we evaluate our model trained with random 100-node problems on real-world TSPLIB instances with 50-200 nodes. The compared baselines include DIFUSCO and baselines listed in [68]'s Table 3. Table 3 shows T2T's superiority achieving merely 0.133% optimality gap, 58.3% better than the best previous method. Detailed results of Table 3 and the results on large-scale TSPLIB instances with 200-1000 nodes in Appendix A. For each instance, we normalize the coordinates to [0,1] and solve with the solving algorithms.

Table 3: Evaluation on TSPLIB dataset.

| Method | Drop |
|---|---|
| AM | 16.767% |
| GCN | 40.035% |
| Learn2OPT | 1.725% |
| GNNGLS | 1.529% |
| DIFUSCO | 0.319% |
| T2T | **0.133%** |

**Validation of Gradient Search with Worse Distribution Learning in Training.** We investigate the performance over $\frac{1}{15}$ amount of training data and the model trained with lower-quality feasible solutions produced by *farthest insertion* heuristic (with 7.50% gap to optimality). Results in Fig. 5 validate the effectiveness of gradient search even if the distribution is poorly learned, suggesting that T2T

Table 4: Generalization results. *Tour length* and *drop* with **Greedy Decoding** are reported.

| Testing | Training | TSP-50 | TSP-100 | TSP-500 | TSP-1000 |
|---|---|---|---|---|---|
| TSP-50 | DIFUSCO | **5.69, 0.09%** | 5.70, 0.25% | 5.83, 2.55% | 5.84, 2.71% |
|  | T2T | **5.69, 0.02%** | 5.70, 0.11% | 5.78, 1.60% | 5.75, 1.10% |
| TSP-100 | DIFUSCO | 7.87, 1.44% | **7.78, 0.23%** | 8.03, 3.44% | 8.02, 3.31% |
|  | T2T | 7.80, 0.55% | **7.77, 0.08%** | 7.95, 2.47% | 7.91, 1.96% |
| TSP-500 | DIFUSCO | 17.31, 4.61% | 17.05, 3.04% | **16.78, 1.40%** | 16.86, 1.85% |
|  | T2T | 17.18, 3.79% | 16.92, 2.25% | **16.68, 0.81%** | 16.72, 1.00% |
| TSP-1000 | DIFUSCO | 24.17, 4.54% | 24.04, 3.98% | 23.65, 2.30% | **23.63, 2.21%** |
|  | T2T | 24.20, 4.66% | 23.85, 3.16% | 23.47, 1.51% | **23.41, 1.23%** |

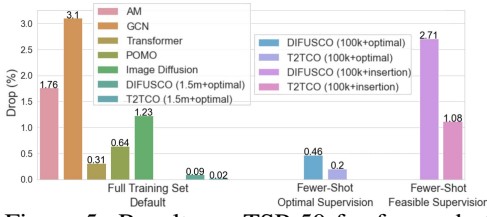

Figure 5: Results on TSP-50 for fewer-shot training with lower-quality supervision.

Table 5: Results on MIS. TS: Tree Search, UL: Unsupervised Learning. DIFUSCO and T2T are compared in the same running settings, while other numbers are quoted from [35; 43].

| ALGORITHM | TYPE | SATLIB | | | ER-[700-800] | | |
|---|---|---|---|---|---|---|---|
|  |  | SIZE↑ | DROP↓ | TIME | SIZE↑ | DROP↓ | TIME |
| KaMIS [69] | Heuristics | 425.96* | – | 37.58m | 44.87* | – | 52.13m |
| Gurobi [66] | Exact | 425.95 | 0.00% | 26.00m | 41.28 | 7.78% | 50.00m |
| Intel [70] | SL+Grdy | 420.66 | 1.48% | 23.05m | 34.86 | 22.31% | 6.06m |
| DIMES [35] | RL+Grdy | 421.24 | 1.11% | 24.17m | 38.24 | 14.78% | 6.12m |
| DIFUSCO [19] | SL+Grdy | 424.56 | 0.33% | 8.25m | 36.55 | 18.53% | 8.82m |
| T2T (Ours) | SL+Grdy | **425.02** | **0.22%** | 8.12m | **39.56** | **11.83%** | 8.53m |
| Intel [70] | SL+TS | – | – | – | 38.80 | 13.43% | 20.00m |
| DGL [71] | SL+TS | – | – | – | 37.26 | 16.96% | 22.71m |
| LwD [72] | RL+S | 422.22 | 0.88% | 18.83m | 41.17 | 8.25% | 6.33m |
| GFlowNets [43] | UL+S | 423.54 | 0.57% | 23.22m | 41.14 | 8.53% | 2.92m |
| DIFUSCO [19] | SL+S | 425.13 | 0.19% | 26.32m | 40.35 | 10.07% | 32.98m |
| T2T (Ours) | SL+S | **425.22** | **0.17%** | 23.80m | **41.37** | **7.81%** | 29.73m |

stands less sensitive to the quality of labels in training than other peer methods. It further demonstrates the generality of T2T especially when high-quality solution labels are not easily available.

## 4.3 Experiments for MIS

**Datasets.** Two datasets are tested for the MIS problem following [70; 72; 71; 35; 19], include SATLIB [73] and Erdős–Rényi (ER) graphs [74]. SATLIB contains SAT instances encoded in DIMACS CNF format, which are then reduced to MIS instances. ER graphs are randomly generated with each edge maintaining a fixed probability of being present or absent, independently of the other edges. We adopt ER graphs of 700 to 800 nodes with the pairwise connection probability set as 0.15.

**Metrics.** Following previous works [12; 33; 35; 19], we adopt three evaluation metrics to measure model performance: 1) Size: the average size of the solutions w.r.t. the corresponding instances, i.e. the objective. 2) Drop: the relative performance drop w.r.t. length compared to the optimal solution or the reference solution; 3) Time: the average computational time required to solve the problems.

**Main Results.** Table 5 presents results. The baselines include state-of-the-art neural methods with greedy decoding and sampling decoding (×4), as well as exact solver Gurobi [66] and heuristic solver KaMIS [69]. The solving time of Gurobi is set as comparable to neural solvers, thus it does not reach optimality. DIFUSCO and T2T are compared in the same conditions. The results of other baselines are quoted from [35]. DIFUSCO and T2T adopt 50 and 20 inference steps, respectively. In the greedy decoding setting, compared to previous neural solvers, T2T improves the performance drop from 0.33% to 0.22% for SATLIB dataset and from 18.53% to 11.83% for ER-[700-800] dataset.

## 5 Conclusion and Future work

We have presented a principled learning paradigm for solving hard-constrained optimization problems specifically combinatorial optimization, which bridges the offline training with reference solution as supervision to the instance-wise solving in testing which calls for more direct gradient feedback from the objective score. Experiments on TSP and MIS show the strong performance of our approach. With the potential generality, further validation across a wider range of CO problems is the immediate further work, including those with general representing capabilities such as Mixed Integer Programming (MIP) and ML-dominant special problems like Quadratic Assignment Problem (QAP). Another promising future direction is to explore the potential of recent (scalable) graph Transformers [75; 76; 77] as expressive encoder backbones for solving complex CO problems.

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

# Appendix

## A    Supplementary Experiments

**Results on TSPLIB.** We evaluate our model trained with random 100-node problems on real-world TSPLIB instances with 50-200 nodes. The compared baselines include DIFUSCO and baselines listed in [68]'s Table 3. Table 6 presents the results and shows T2T's superiority achieving merely 0.133% optimality gap, 58.3% better than the best previous method. We also supplement the results (optimality drop) of DIFUSCO and T2T on large-scale TSPLIB benchmark instances with 200-1000 nodes, as shown in Table 7. For each instance, we normalize the coordinates to [0,1] and solve with the solving algorithms.

Table 6: Solution quality for methods trained on random 100-node problems and evaluated on **TSPLIB instances** with 50-200 nodes. The comparison includes baselines in [68]'s Table 3. Results of DIFUSCO and T2T are based on $4\times$ sampling decoding with 2OPT post-processing. Results of other baselines are from [68].

| INSTANCES | AM | GCN | Learn2OPT | GNNGLS | DIFUSCO | T2T |
|---|---|---|---|---|---|---|
| eil51 | 16.767% | 40.025% | 1.725% | 1.529% | 0.314% | 0.314% |
| berlin52 | 4.169% | 33.225% | 0.449% | 0.142% | 0.000% | 0.000% |
| st70 | 1.737% | 24.785% | 0.040% | 0.764% | 0.172% | 0.000% |
| eil76 | 1.992% | 27.411% | 0.096% | 0.163% | 0.217% | 0.163% |
| pr76 | 0.816% | 27.793% | 1.228% | 0.039% | 0.043% | 0.039% |
| rat99 | 2.645% | 17.633% | 0.123% | 0.550% | 0.016% | 0.000% |
| kroA100 | 4.017% | 28.828% | 18.313% | 0.728% | 0.050% | 0.000% |
| kroB100 | 5.142% | 34.686% | 1.119% | 0.147% | 0.000% | 0.000% |
| kroC100 | 0.972% | 35.506% | 0.349% | 1.571% | 0.000% | 0.000% |
| kroD100 | 2.717% | 38.018% | 0.866% | 0.572% | 0.000% | 0.000% |
| kroE100 | 1.470% | 26.589% | 1.832% | 1.216% | 0.000% | 0.000% |
| rd100 | 3.407% | 50.432% | 1.725% | 0.003% | 0.000% | 0.000% |
| eil101 | 2.994% | 26.701% | 0.387% | 1.529% | 0.124% | 0.000% |
| lin105 | 1.739% | 34.902% | 1.867% | 0.606% | 0.441% | 0.393% |
| pr107 | 3.933% | 80.564% | 0.898% | 0.439% | 0.714% | 0.155% |
| pr124 | 3.677% | 70.146% | 10.322% | 0.755% | 0.997% | 0.584% |
| bier127 | 5.908% | 45.561% | 3.044% | 1.948% | 1.064% | 0.718% |
| ch130 | 3.182% | 39.090% | 0.709% | 3.519% | 0.077% | 0.077% |
| pr136 | 5.064% | 58.673% | 0.000% | 3.387% | 0.182% | 0.000% |
| pr144 | 7.641% | 55.837% | 1.526% | 3.581% | 1.816% | 0.000% |
| ch150 | 4.584% | 49.743% | 0.312% | 2.113% | 0.473% | 0.324% |
| kroA150 | 3.784% | 45.411% | 0.724% | 2.984% | 0.193% | 0.193% |
| kroB150 | 2.437% | 56.745% | 0.886% | 3.258% | 0.366% | 0.021% |
| pr152 | 7.494% | 33.925% | 0.029% | 3.119% | 0.687% | 0.687% |
| u159 | 7.551% | 38.338% | 0.054% | 1.020% | 0.000% | 0.000% |
| rat195 | 6.893% | 24.968% | 0.743% | 1.666% | 0.887% | 0.018% |
| d198 | 373.020% | 62.351% | 0.522% | 4.772% | 0.000% | 0.000% |
| kroA200 | 7.106% | 40.885% | 1.441% | 2.029% | 0.259% | 0.000% |
| kroB200 | 8.541% | 43.643% | 2.064% | 2.589% | 0.171% | 0.171% |
| Mean | 16.767% | 40.025% | 1.725% | 1.529% | 0.319% | **0.133%** |

**Generalization Results on Distribution Shift.** The generalization results of the diffusion-based methods (e.g., DIFUSCO and T2T) on the distribution (e.g., cluster, gaussian, etc., instead of uniform) are presented in Table. 8 and Table. 9. The TSP data of different distributions are from [78].

**Effect of Number of Iterations.** Table. 10 shows the results of T2T with varying #iterations. Each iteration contributes to a single process of solution improvement, and employing more iterations can lead to improved solutions. However, too many iterations could lead to additional computational overhead. As a result, we aim to strike a balance in this trade-off and set #iterations to 3.

**Effect of Graph Sparsification Hyperparameter $k$.** Table. 10 shows the results of T2T with varying $k$. A larger value of $k$ can yield improved results, yet it can also lead to increased computational overhead. Thus, we strike a balance in this trade-off and set $k$ to 50.

Table 7: Solution quality for methods trained on random 500-node problems and evaluated on **TSPLIB instances** with 200-1000 nodes. Results of DIFUSCO and T2T are based on $4\times$ sampling decoding with 2OPT post-processing.

| INSTANCES | DIFUSCO | T2T | INSTANCES | DIFUSCO | T2T |
|---|---|---|---|---|---|
| pr439 | 1.28% | 0.26% | d493 | 0.00% | 0.00% |
| pcb442 | 0.00% | 0.00% | p654 | 0.00% | 0.00% |
| fl417 | 0.00% | 0.00% | d657 | 0.00% | 0.00% |
| tsp225 | 0.99% | 0.59% | rat783 | 1.67% | 1.22% |
| u724 | 0.00% | 0.00% | ts225 | 0.32% | 0.32% |
| pr299 | 0.57% | 0.14% | lin318 | 0.30% | 0.30% |
| rd400 | 0.12% | 0.00% | u574 | 0.00% | 0.00% |
| rat575 | 0.83% | 0.26% | pr1002 | 2.21% | 1.42% |
| a280 | 1.11% | 0.00% | pr226 | 0.34% | 0.22% |
| pr264 | 1.12% | 0.48% | **Mean** | **0.57%** | **0.27%** |

Table 8: Generalization results from the model trained with Uniform-100 data.

| Methods | Decoding | Uniform-100 | Gaussian-100 |
|---|---|---|---|
| DIFUSCO | Greedy | 7.78, 0.20% | 5.72, 0.72% |
| T2T | Greedy | 7.76, 0.06% | 5.69, 0.28% |
| DIFUSCO | Sampling | 7.76, 0.00% | 5.68, 0.09% |
| T2T | Sampling | 7.76, 0.00% | 5.68, 0.02% |

# B   Applicability of T2T

We provide a summary concerning the types of problems that T2T can effectively manage:

- Mandatory Requirements:
  - The decision variables are limited to a countable and finite set of values. Currently, we focus on binary decision variables and the model predicts a heatmap for the solution. But it can be easily extended to prediction through the multinomial diffusion. Note that even binary variables can cover a wide range of CO problems especially on graphs, since most problems can be attributed to edge-selecting problems like TSP, CVRP, and node-selecting problems like MIS, MAX Clique.
  - A problem-specific post-processing procedure is required to transform the prediction into feasible solutions. For problems featuring binary decision variables, this prerequisite can be easily met with the greedy decoding strategy, which sequentially inserts variables (edges or nodes) with the highest confidence if there are no conflicts.

- Nice-to-have Requirements:
  - The constraints can be transformed to some sort of penalty with meaningful gradient. While it is preferable to include a constraint penalty term in Eq. 7, the utilization of the optimization objective alone is also effective (as evidenced by the TSP results). This is due to the diffusion's capacity to implicitly learn constraints.

# C   Proofs

**Proposition 1.** *The optimization-enforced denoising probability estimation $p_\theta(\mathbf{x}_{t-1}|\mathbf{x}_t, G, y^*)$ equals to $Zp_\theta(\mathbf{x}_{t-1}|\mathbf{x}_t, G)p(y^*|\mathbf{x}_{t-1}, G)$, where $Z$ is a normalizing constant.*

*Proof.* Since $p_\theta(\mathbf{x}_{t-1}|\mathbf{x}_t, G, y^*)$ estimates $q(\mathbf{x}_{t-1}|\mathbf{x}_t, G, y^*)$, we aim to show $q(\mathbf{x}_{t-1}|\mathbf{x}_t, G, y^*) \propto q(\mathbf{x}_{t-1}|\mathbf{x}_t, G)p(y^*|\mathbf{x}_{t-1}, G)$, such that as $q(\mathbf{x}_{t-1}|\mathbf{x}_t, G)$ can be readily estimated by $p_\theta(\mathbf{x}_{t-1}|\mathbf{x}_t, G)$, $q(\mathbf{x}_{t-1}|\mathbf{x}_t, G, y^*)$'s neural estimation $p_\theta(\mathbf{x}_{t-1}|\mathbf{x}_t, G, y^*)$ can be achieved by $Zp_\theta(\mathbf{x}_{t-1}|\mathbf{x}_t, G)p(y^*|\mathbf{x}_{t-1}, G)$, where $Z$ is a normalizing constant.

Table 9: Generalization results from the model trained with Uniform-500 data.

| Methods | Decoding | Uniform-500 | Rotation-500 | Explosion-500 | Cluster$_3^{10}$-300 | Cluster$_7^{50}$-300 |
|---|---|---|---|---|---|---|
| DIFUSCO | Greedy | 16.82, 1.63% | 12.69, 2.44% | 12.06, 2.80% | 9.72, 2.64% | 5.80, 3.09% |
| T2T | Greedy | 16.68, 0.78% | 12.51, 1.00% | 11.94, 1.76% | 9.61, 1.49% | 5.75, 2.17% |
| DIFUSCO | Sampling | 16.68, 0.81% | 12.60, 1.65% | 11.94, 1.78% | 9.62, 1.57% | 5.75, 2.18% |
| T2T | Sampling | 16,62, 0.42% | 12.45, 0.48% | 11.87, 1.18% | 9.55, 0.85% | 5.72, 1.67% |

Table 10: TSP-500 Greedy results of the hyperparameter #iterations.

| #iterations | 0 | 1 | 3 | 5 | 10 |
|---|---|---|---|---|---|
| Tour Length↓ | 16.813 | 16.705 | 16.683 | 16.678 | 16.669 |
| Drop↓ | 1.59% | 0.94% | 0.80% | 0.77% | 0.72% |

We first show that $p(y^*|\mathbf{x}_{t-1}, G)$ does not depend on $\mathbf{x}_t$:

$$p(y^*|\mathbf{x}_{t-1}, \mathbf{x}_t, G) = q(\mathbf{x}_t|\mathbf{x}_{t-1}, y^*, G)\frac{p(y^*|\mathbf{x}_{t-1}, G)}{q(\mathbf{x}_t|\mathbf{x}_{t-1}, G)} \tag{14}$$

$$= q(\mathbf{x}_t|\mathbf{x}_{t-1})\frac{p(y^*|\mathbf{x}_{t-1}, G)}{q(\mathbf{x}_t|\mathbf{x}_{t-1})} \tag{15}$$

$$= p(y^*|\mathbf{x}_{t-1}, G) \tag{16}$$

Then the probability distribution $q(\mathbf{x}_{t-1}|\mathbf{x}_t, G, y^*)$ can be derived by:

$$q(\mathbf{x}_{t-1}|\mathbf{x}_t, G, y^*) = \frac{q(\mathbf{x}_{t-1}, \mathbf{x}_t, G, y^*)}{q(\mathbf{x}_t, G, y^*)} \tag{17}$$

$$= \frac{q(\mathbf{x}_{t-1}, \mathbf{x}_t, G, y^*)}{q(y^*|\mathbf{x}_t, G)q(\mathbf{x}_t, G)} \tag{18}$$

$$= \frac{q(y^*|\mathbf{x}_{t-1}, \mathbf{x}_t, G)q(\mathbf{x}_{t-1}|\mathbf{x}_t, G)q(\mathbf{x}_t, G)}{q(y^*|\mathbf{x}_t, G)q(\mathbf{x}_t, G)} \tag{19}$$

$$= \frac{q(y^*|\mathbf{x}_{t-1}, G)q(\mathbf{x}_{t-1}|\mathbf{x}_t, G)}{q(y^*|\mathbf{x}_t, G)} \tag{20}$$

$q(y^*|\mathbf{x}_t, G)$ is not depend on $\mathbf{x}_{t-1}$, thus it can be regarded as a constant. Now we can obtain $q(\mathbf{x}_{t-1}|\mathbf{x}_t, G, y^*) \propto q(\mathbf{x}_{t-1}|\mathbf{x}_t, G)p(y^*|\mathbf{x}_{t-1}, G)$, completing the proof.

□

## D    Discussion from Another Aspect: Constructive and Improvement Solvers

Existing CO solvers including both traditional and emerging ML-based methods, generally can be categorized into two groups as summarized in detail in Table 12. Constructive solvers solve the problem from scratch in one shot or multiple steps, and the algorithm ends once a complete solution is obtained. However, effectively enhancing the final solution quality (e.g. the objective score) while ensuring the constraints can be challenging for constructive solvers due to the absence of explicit reference to complete solutions (as the constraints are often obeyed globally across the dimensions of the solution). While in contrast, improvement solvers yield and maintain a series of complete solutions during the improvement procedure to optimize (namely minimize) the objective in a score decreasing trend (with fluctuation). Compared with constructive solvers, improvement (search-based) solvers [36; 37; 5; 37; 7; 38] seek an enhanced solution by iteratively performing local search within its neighborhood. The algorithm pipelines generally involve selecting specific regions and locally optimizing sub-solutions. This paradigm often offers better tractability and a more direct correlation to objective score thanks to the explicit involvement of the objective and accessibility of complete solutions. While improvement-based solvers dominate in traditional heuristics (e.g., LKH [60; 63]), for learning class methods, the constructive solvers remain relatively major in number, given their more direct compatibility with network prediction tasks.

For constructive ML-solvers, the training procedures can be classified into reinforcement learning (RL) and supervised learning (SL). RL methods [12; 6; 8] try to directly optimize the objective

Table 11: TSP-500 Greedy results of the graph sparsification hyperparameter $k$.

| $k$ | 10 | 50 | 100 | 200 |
|---|---|---|---|---|
| Tour Length↓ | 16.687 | 16.681 | 16.678 | 16.674 |
| Drop↓ | 0.83% | 0.79% | 0.77% | 0.74% |

Table 12: Constructive and improvement solvers. ***Sol. Dist.*:** solution distribution learning capability; ***LS*:** using local search scheme. A constructive solver can be combined with a search-based solver, where the latter is responsible for the search component being not part of the constructive approach.

| Neural solver | Paradigm | Training | *Sol. Dist.*? | *LS*? |
|---|---|---|---|---|
| AM [12] | Constructive | Reinforce | No | No |
| POMO [6] | Constructive | Reinforce | No | No |
| Sym-NCO [8] | Constructive | Reinforce | No | No |
| GCN [33] | Constructive | Supervised | No | No |
| DIMES [35] | Constructive | Reinforce | No | No |
| NeuRewriter [5] | Improvement | Reinforce | No | Yes |
| Learning 2OPT [36] | Improvement | Reinforce | No | Yes |
| Learn2delegate [7] | Improvement | Reinforce | No | Yes |
| TAM [38] | Improvement | Reinforce | No | Yes |
| CVAE-Opt [23] | Constructive | Supervised | Yes | No |
| NeuRouter [28] | Constructive | Supervised | Yes | No |
| DIFUSCO [19] | Constructive | Supervised | Yes | No |
| T2T (ours) | Improvement | Supervised | Yes | Yes |

function, but since a complete solution is not obtained until the end, the intermediate process lacks a valid reference. SL methods [33; 35; 19] typically fit the network prediction to the reference solutions. The lack of objective awareness and local search procedures limits their ability to surpass the supervised solvers. The training procedures of current improvement ML-solvers are based on RL.

Generative modeling for CO [23; 19] currently falls into a subclass of constructive solvers. Generative modeling for CO currently falls into a subclass of constructive solvers, which leverages the powerful expressive and modeling capabilities of generative models, e.g., VAE [23] and the diffusion model [19]. However, as mentioned above the model lacks awareness of the optimization objective and does not introduce a search procedure to fully utilize the estimated solution distribution. In contrast to the constructive way for CO problem solving, our T2T serves as an improvement solver the strong compatibility between the diffusion model and the diffuse-and-denoise local search paradigm with objective feedback in the loop. In our approach, generative diffusion model and local search are both used (which is the first time to our best knowledge), in a complementary and synergistic way to achieve state-of-the-art performance in both objective score and solving time. Compared to existing improvement solvers, such a design facilitates parallel optimization across multiple sub-regions of the entire solution, greatly improving efficiency while also eliminating the need for the iterative selection of optimization regions.

## E  Network Architecture Details

### E.1  Input Embedding Layer

Given node vector $x \in \mathbb{R}^{N \times 2}$, weighted edge vector $e \in \mathbb{R}^E$, denoising timestep $t \in \{\tau_1, \ldots, \tau_M\}$, where $N$ denotes the number of nodes in the graph, and $E$ denotes the number of edges, we compute the sinusoidal features of each input element respectively:

$$\tilde{x}_i = \text{concat}(\tilde{x}_{i,0}, \tilde{x}_{i,1}) \tag{21}$$

$$\tilde{x}_{i,j} = \text{concat}\left(\sin\frac{x_{i,j}}{T^{\frac{0}{d}}}, \cos\frac{x_{i,j}}{T^{\frac{0}{d}}}, \sin\frac{x_{i,j}}{T^{\frac{2}{d}}}, \cos\frac{x_{i,j}}{T^{\frac{2}{d}}}, \ldots, \sin\frac{x_{i,j}}{T^{\frac{d}{d}}}, \cos\frac{x_{i,j}}{T^{\frac{d}{d}}}\right) \tag{22}$$

$$\tilde{e}_i = \text{concat}\left(\sin\frac{e_i}{T^{\frac{0}{d}}}, \cos\frac{e_i}{T^{\frac{0}{d}}}, \sin\frac{e_i}{T^{\frac{2}{d}}}, \cos\frac{e_i}{T^{\frac{2}{d}}}, \ldots, \sin\frac{e_i}{T^{\frac{d}{d}}}, \cos\frac{e_i}{T^{\frac{d}{d}}}\right) \tag{23}$$

$$\tilde{t} = \text{concat}\left(\sin\frac{t}{T^{\frac{0}{d}}}, \cos\frac{t}{T^{\frac{0}{d}}}, \sin\frac{t}{T^{\frac{2}{d}}}, \cos\frac{t}{T^{\frac{2}{d}}}, \ldots, \sin\frac{t}{T^{\frac{d}{d}}}, \cos\frac{t}{T^{\frac{d}{d}}}\right) \tag{24}$$

where $d$ is the embedding dimension, $T$ is a large number (usually selected as 10000), $\text{concat}(\cdot)$ denotes concatenation.

Next, we compute the input features of the graph convolution layer:

$$x_i^0 = W_1^0 \tilde{x}_i \tag{25}$$

$$e_i^0 = W_2^0 \tilde{e}_i \tag{26}$$

$$t^0 = W_4^0(\text{ReLU}(W_3^0 \tilde{t})) \tag{27}$$

where $t^0 \in \mathbb{R}^{d_t}$, $d_t$ is the time feature embedding dimension. Specifically, for TSP, the embedding input edge vector $e$ is a weighted adjacency matrix, which represents the distance between different nodes, and $e^0$ is computed as above. For MIS, we initialize $e^0$ to a zero matrix $0^{E \times d}$.

## E.2 Graph Convolution Layer

Following [33], the cross-layer convolution operation is formulated as:

$$x_i^{l+1} = x_i^l + \text{ReLU}(\text{BN}(W_1^l x_i^l + \sum_{j \sim i} \eta_{ij}^l \odot W_2^l x_j^l)) \tag{28}$$

$$e_{ij}^{l+1} = e_i^l + \text{ReLU}(\text{BN}(W_3^l e_{ij}^l + W_4^l x_i^l + W_5^l x_j^l)) \tag{29}$$

$$\eta_{ij}^l = \frac{\sigma(e_{ij}^l)}{\sum_{j' \sim i} \sigma(e_{ij'}^l) + \epsilon} \tag{30}$$

where $x_i^l$ and $e_{ij}^l$ denote the node feature vector and edge feature vector at layer $l$, $W_1, \cdots, W_5 \in \mathbb{R}^{h \times h}$ denote the model weights, $\eta_{ij}^l$ denotes the dense attention map. The convolution operation integrates the edge feature to accommodate the significance of edges in routing problems.

For TSP, we aggregate the timestep feature with the edge convolutional feature and reformulate the update for edge features as follows:

$$e_{ij}^{l+1} = e_{ij}^l + \text{ReLU}(\text{BN}(W_3^l e_{ij}^l + W_4^l x_i^l + W_5^l x_j^l)) + W_6^l(\text{ReLU}(t^0)) \tag{31}$$

For MIS, we aggregate the timestep feature with the node convolutional feature and reformulate the update for node features as follows:

$$x_i^{l+1} = x_i^l + \text{ReLU}(\text{BN}(W_1^l x_i^l + \sum_{j \sim i} \eta_{ij}^l \odot W_2^l x_j^l)) + W_6^l(\text{ReLU}(t^0)) \tag{32}$$

## E.3 Output Layer

The prediction of the edge heatmap in TSP and node heatmap in MIS is as follows:

$$e_{i,j} = \text{Softmax}(\text{norm}(\text{ReLU}(W_e e_{i,j}^L))) \tag{33}$$

$$x_i = \text{Softmax}(\text{norm}(\text{ReLU}(W_n x_i^L))) \tag{34}$$

where $L$ is the number of GCN layers and $\text{norm}$ is layer normalization.

## E.4 Hyper-parameters

For both TSP and MIS tasks, we construct a 12-layer GCN derived above. We set the node, edge, and timestep embedding dimension $d = 256, 128$ for TSP and MIS tasks, respectively.

# F Fast Inference for Discrete Diffusion Models

The technique of denoising diffusion implicit models (DDIMs) [49] is introduced for accelerating inference and solution reconstruction. Let $x_{1:T}$ denotes the latent variables and $\{\tau_1, \ldots, \tau_M\}$ is one increasing subset of time sequences $\{1, \ldots, T\}$. $\tau_M = T$ for the inference processes and $\alpha T$ for the reconstruction processes. The diffusion processes can be generated to a class non-Markovian processes with the same training objective, where the inference processes traverse only on a subset of latent variables. The inference processes are reformulated as:

$$
\begin{aligned}
q(\mathbf{x}_{\tau-1}|\mathbf{x}_\tau, \mathbf{x}_0) &= \frac{q(\mathbf{x}_\tau|\mathbf{x}_{\tau-1}, \mathbf{x}_0)q(\mathbf{x}_{\tau-1}|\mathbf{x}_0)}{q(\mathbf{x}_\tau|\mathbf{x}_0)} \\
&= \mathrm{Cat}\left(\mathbf{x}_{\tau-1}; \mathbf{p} = \frac{\mathbf{x}_\tau \mathbf{Q}_\tau^\top \odot \mathbf{x}_0 \overline{\mathbf{Q}}_{\tau-1}}{\mathbf{x}_0 \overline{\mathbf{Q}}_\tau \mathbf{x}_\tau^\top}\right)
\end{aligned} \tag{35}
$$

For both TSP and MIS tasks, we adopt the cosine denosing scheduler such that $\tau_i = \lfloor \cos\left(\frac{1-\pi \cdot ci}{2}\right) \cdot T \rfloor$ for some constant $c$.

# G Experimental Details

## G.1 Training Resource Requirement

Table 13: Details about the training resource requirement of T2T framework. The results are calculated on A100 GPUs.

We provide the details about the offline training resource requirement of T2T framework in Table 13. The results are calculated on A100 GPUs. For comparison reference: AM [12] requires 128M instances generated on the fly for training TSP-100, and it requires 45.8h on 2 1080Ti GPUs; POMO [6] requires 200M instances generated on the fly for training TSP-100, and it requires 1 week on a single Titan RTX; Sym-NCO [8] based on POMO requires 2 weeks on a single A100, Sym-NCO [8] based on AM requires 3 days on 4 A100 GPUs.

| Problem Scale | Dataset Size | 1 GPU | 2 GPUs | 4 GPUs | GPU Mem |
|---|---|---|---|---|---|
| TSP-50 | 1,502 k | 56h 24m | 30h 49m | 19h 8m | 13.2 GB |
| TSP-100 | 1,502 k | 206h 20m | 109h 45m | 59h 55m | 24.0 GB |
| TSP-500 | 128 k | 64h 26m | 34h 37m | 22h 46m | 19.1 GB |
| TSP-1000 | 64 k | 123h 31m | 65h 46m | 38h 35m | 35.8 GB |

## G.2 Hyperparamters

We conduct experiments on TSP and MIS benchmarks with our methods and compare the performance with SOTA learning-based solvers, heuristics, and exact solvers. We select hyperparameters to balance the inference time and the solution quality. Specifically, the inference time of our method is approximate with other learning-based frameworks. For each benchmark, we adopt 20 inference steps and 10 objective-guided denoising steps for reconstruction, which involves 3 iterations and selects the optimal solutions. The disruption ratio $\alpha$ associated with each benchmark is listed in Table. 14.

Table 14: Disruption Ratio for each benchmark.

| Benchmark | TSP-50 | TSP-100 | TSP-500 | TSP-1000 | SATLIB | ER-[700-800] |
|---|---|---|---|---|---|---|
| $\alpha$ | 0.25 | 0.25 | 0.4 | 0.4 | 0.1 | 0.1 |

### G.3 Baseline Settings

#### G.3.1 TSP Benchmarks

**TSP-50/100.** For TSP-50 and TSP-100, we compare our proposed T2T with 10 baselines, including one exact solver Concorde [62], two heuristic solvers 2OPT [64] and Farthest Insertion, and 7 learning-based solvers, i.e., AM [12], GCN [33], Transformer [65], POMO [6], Sym-NCO [8], Image Diffusion [59] and DIFUSCO [19]. The post-processing includes greedy sampling and 2OPT refinement. To ensure a fair comparison with similar runtimes, we set the number of inference steps for DIFUSCO to 120.

**TSP-500/1000.** For TSP-500 and TSP-1000, we compare our method with 2 exact solvers Concorde [62] and Gurobi [66], 2 heuritics solvers LKH-3 [63] and Farthest Insertion, and 6 learning-based methods including EAN [67], AM [12], GCN [33], POMO+EAS [17], DIMES [35] and DIFUSCO [19]. The learning-based methods can further be categorized into supervised learning (SL) and reinforcement learning (RL). The post-processing includes greedy sampling (Grdy, G), multiple sampling (S), 2OPT refinement (2OPT), beam search (BS), active search (AS), and their combinations. To ensure a fair comparison with similar runtimes, we set the number of inference steps for DIFUSCO to 120.

#### G.3.2 MIS Benchmarks

We evaluate our method on two benchmarks, namely SATLIB and ER-[700-800]. For both benchmarks, we compare T2T with one exact solver Gurobi [66], one heuristic solver KaMIS [69] and 5 learning-based frameworks Intel [70], DGL [70], LwD [72], DIMES [35] and DIFUSCO [19]. The post-processing includes greedy sampling (Grdy) and tree search (TS). Specifically, we set the inference steps 50 for DIFUSCO on both benchmarks.

### G.4 Graph Sparsification

In TSP, the number of edges in the graph increases quadratically with the number of nodes, which poses scalability challenges. To address this, we adopt graph sparsification by limiting each node to connect only with its k nearest neighbors (KNN) based on Euclidean distance. Specifically, we set $k = 50$ for TSP-500 and TSP-1000.

### G.5 Visualization

Fig. 6 visualizes the specific process of T2T with 5 rounds of gradient search on TSP-50.

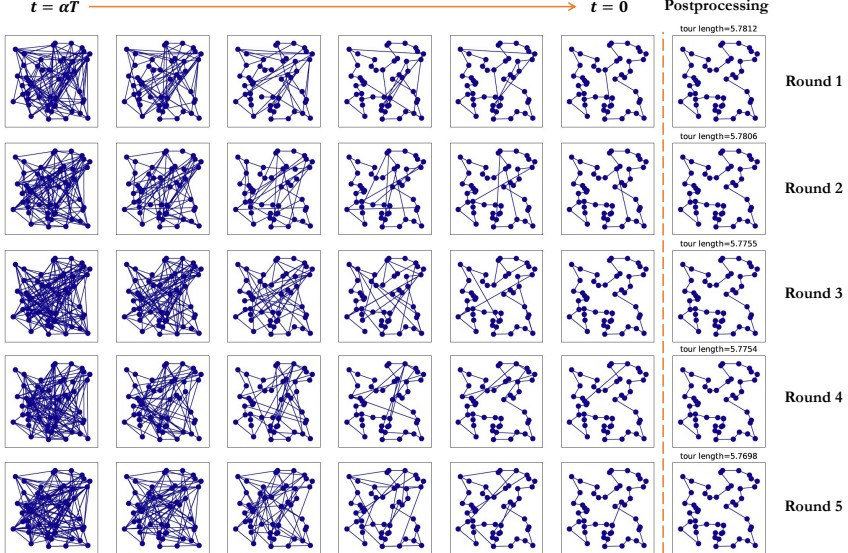

Figure 6: The visualization of T2T solving TSP-50, as corresponding to the orange part in Fig. 3.

