# OpenReview forum: "T2T: From Distribution Learning in Training to Gradient Search in Testing for Combinatorial Optimization"
_NeurIPS.cc/2023/Conference — NeurIPS 2023 poster_

### Official Review · Reviewer_gU16 · 2023-06-15

**Soundness:** 3 good
**Presentation:** 3 good
**Contribution:** 2 fair
**Rating:** 6
**Confidence:** 3

**Summary:**

This paper proposes a novel framework, T2TCO, for solving combinatorial optimization problems. Specifically, it trains a generative model (i.e., discrete diffusion solver from [1]) to estimate the distribution of high-quality solutions for each instance, and, during testing, conducts an effective local search (i.e., iterative objective-aware denoising) guided by the gradient feedback within the estimated solution space. The experiments are conducted on TSP and MIS to demonstrate the superiority over previous SOTA methods.

**Strengths:**

* The writing of this paper is good. The motivation of the proposed method is clear. The drawbacks of previous tailored search methods in the test phase (i.e., active search or meta-learning needs to update model parameters), and that of the constructive diffusion solver [1] (i.e., without explicitly considering the optimization of CO objective during denoising) are discussed clearly.

* The proposed method seems to be sound and novel. This paper uses the discrete diffusion model from [1] to model the high-quality solution distribution for each instance. The novelty of the proposed T2TCO could be mainly attributed to the objective-aware denoising process $p_{\theta}(x_{t-1}|x_t,G,y^*)$ with gradient feedback (see Section 3.3.1), along with its theoretical analyses.

* The empirical results are good, achieving SOTA on TSP and MIS.

**Weaknesses:**

* The proposed method is only applicable to diffusion-based methods [1], which may limit the scope of this paper. Moreover, it is an incremental work upon DIFUSCO [1].
* Computational Complexity:
  * The training cost of the diffusion model is extremely high, requiring **four A100 GPUs**.  It would be an interesting direction on improving the efficiency of the diffusion-based models for COPs.
  * Could you provide further details about the needed training time (if training on a single GPU) and GPU memory?
  * How about the training and inference complexity compared to DIFUSCO [1] (e.g., given the same number of sampling steps)?
* Empirical Evaluation:
  * How about the performance if using Monte-Carlo Tree Search (MCTS) [2] as the post-inference decoding?
  * Any training setups for baselines? POMO [3] could achieve 0.03\% and 0.14\% on TSP50 and TSP100, respectively, while you report 0.64\% and 1.07\% in Table 1. These differences are not negligible.
  * How about the generalization performance? DIFUSCO [1] has conducted the generalization study in their empirical analyses. Moreover, recent works [4, 5] consider the generalization study of neural (TSP) solvers across both size and distribution. It is interesting to see how your method performs in this setting.
  * The detailed results on the classical benchmark dataset (e.g., TSPLIB [6]) are needed for both T2TCO and DIFUSCO.
* Minor:
  * In line 96, "autoregressive" -> "construction".
  * In line 113, ",T2TCO" -> ", T2TCO".
  * In line 170, "$Cat$" -> "Cat".
  * In line 220, " Note" -> "Note"; "to" -> "the".

[1] DIFUSCO: Graph-based diffusion solvers for combinatorial optimization. In arXiv:2302.08224.

[2] Generalize a small pre-trained model to arbitrarily large tsp instances. In AAAI 2021.

[3] POMO: Policy optimization with multiple optima for reinforcement learning. In NeurIPS 2020.

[4] On the generalization of neural combinatorial optimization heuristics. In ECMLPKDD 2022.

[5] Towards Omni-generalizable Neural Methods for Vehicle Routing Problems. In ICML 2023.

[6] http://comopt.ifi.uni-heidelberg.de/software/TSPLIB95

*Note to AC: I am not an expert in diffusion and hence cannot completely ensure the correctness of the equations in Section 3.3.1 and the proof in the Appendix.*

----

**Overall**, despite the limited scope and expensive training cost, the proposed method is sound and novel, which may bring new insights to the ML4CO or NCO community. The empirical results are good as well. Therefore, I lean towards borderline acceptance.

**Questions:**

* Do you plan to release the source code during the rebuttal period?
* Has a similar idea (to the one in Section 3.3.1) been proposed in other domains (e.g., computer vision)?
* Could you discuss further the objective function in Eqs. (6)-(7)? For example, why not use Eq. (1) in [1]?
* Why the energy function is designed as Eq. (10)?
* In Eq. (12), why $-\log\nabla_{x_t}Z$ is removed? It seems that $Z$ is also a function of $x_t$.
* In line 224, is the negative sign missing in the exponentiation, i.e., $\exp(-\cdots)$?
* The authors mention that $x_{t-1}$ is not accessible at step $t$. However, the approximation of $p(y^*|x_{t-1},G)$ still depends on $x_{t-1}$ (in line 224). I'm confused about it. Could you explain it further?

**Limitations:**

Limited scope and training efficiency.

No negative societal impact.

---

> ### Author Rebuttal · Authors · 2023-08-09
>
> Thanks for the thorough review and valuable comments. We are encouraged with your acknowledgment of our writing, motivation, methodology, empirical results, and contribution to the ML4CO community. Below we respond to your specific comments.
>
> **Q1: The proposed method is only applicable to diffusion-based methods, which may limit the scope of this paper.**
>
> Indeed the proposed method is technically developed on top of the diffusion models. However, it is important to note that this technical foundation does not confine its range of applications. For instance, the proposed framework exhibits the capacity to address CO problems whose decision variables are limited to a countable and finite set of values. This versatility encompasses a variety of graph CO problems, spanning both edge-selection problems like TSP and CVRP, as well as node-selecting problems like MIS and MAX Clique.
>
>
> **Q2: More details about the needed training time. The inference complexity compared to DIFUSCO.**
>
> We have provided the details about the offline training resource requirement of T2TCO framework in the general response.
>
> For the comparison to DIFUSCO, the proposed objective-guided denoising step requires ~ 3 times the calculation of the original denoising step. This increase is due to the additional gradient calculations needed to incorporate objective optimization information. However, T2TCO can achieve satisfactory results with fewer steps. The experimental comparisons are based on results with approximately equal computational overhead.
>
>
> **Q3: How about the generalization performance?**
>
> Thanks for the comment. We have supplemented the generalization results in the general response.
>
> **Q4: POMO could achieve 0.03% and 0.14% on TSP50 and TSP100, respectively, while you report 0.64% and 1.07% in Table 1.**
>
> Please note that the results in Table 1 (original) are based on greedy decoding. In this setting, the results are consistent with POMO's Table 2.
>
> **Q5: Experiments on real-world TSPLIB benchmark.**
>
> Thanks for the advice. We have supplemented the experimental results on TSPLIB in the general response.
>
> **Q6: Has a similar idea (to the one in Section 3.3.1) been proposed in other domains (e.g., computer vision)?**
>
> As mentioned in Line 124-126, classifier guidance and classifier-free guidance similarly attempt to introduce additional shifts to the original denoising steps to achieve conditional generation. However, these shifts originate either from an additional predictive classifier or from the difference between the conditional model and the non-conditional model, without a explicit optimization objective.
>
> **Q7: For Eqs. 6-7, why not use Eq. 1 in DIFUSCO as the objective?**
>
> Eq. 1 in DIFUSCO using our notations: $l(x;G)=f(x;G)+valid(x,G)$, where $f(\cdot)$ is the objective function defined on the feasible space and $valid(\cdot)$ is the validation function that returns 0 for feasible solutions and $+\infty$ for invalid solutions.
>
> This objective formulation is general and classical in CO, but we cannot directly utilize it because:
> 1. The need for the approximate objective estimation for the relaxed (infeasible) $x_t$ at time step $t$ requires a relaxed version of $f(\cdot)$ like $f_r(\cdot)$ in Eq. 7.
> 2. The acquisition of gradient feedback from the objective is a requisite, yet the function $valid(\cdot)$ lacks differentiability. Thus we define the differentiable constraint penalty $g_r(\cdot)$ to reflect the extent of constraint violation and transmit reliable gradient signals.
>
>
> **Q8: Why the energy function is designed as Eq. 10?**
>
> Recall that an energy function measures how well the outputs fit the inputs. Eq. 10 aims to quantify the consistency between the inputs $(x_t, G)$ and the output $y$. Such that in the derived distribution (Eq. 7), the best $y$ matching the inputs can maintain the highest probability density, and the probability density is positively correlated with the matching degree.
>
> **Q9: In Eq. 12, why is $-\log\nabla_{x_t}Z$ removed?**
>
> In the distribution defined by Eq. 11, different values of $x_t$ only affect the center position (i.e., $l(x_t;G)$) of this unimodal distribution. Changing $x_t$ only results in shifting this distribution, and therefore, $Z$ as the integral over the entire distribution is not affected by $x_t$ and can be removed in Eq. 12.
>
> **Q10: In line 224, is the negative sign missing in the exponentiation?**
>
> Yes, the negative sign is missing. Sorry for our typo.
>
> **Q11: The authors mention that $x_{t-1}$ is not accessible at step $t$. However, the approximation of $p(y^\*|x_{t-1}, G)$ still depends on $x_{t-1}$ (in line 224).**
>
> In line 224, the expression $[-\nabla l(x_t;G)]^\top x_{t-1}$ indicates how to associate the calculated $[-\nabla l(x_t;G)]^\top$ with $x_{t-1}$. For $p_\theta(x_{t-1}|x_t,G,y^*)=Zp_\theta(x_{t-1}|x_t,G)p(y^*|x_{t-1},G)$, the former term produce the probability of $x_{t-1}$ and the latter term calculates $[-\nabla l(x_t;G)]^\top$. Their combination yields the ultimate output prediction.
>
> **Q12: How about the performance if using Monte-Carlo Tree Search (MCTS) as the post-inference decoding?**
>
> We have made efforts with MCTS, but the diffusion-based models consistently fall short in yielding satisfactory results when coupled with MCTS. This may be due to that the solutions produced by the models are close to feasible rather than soft heatmaps, which might hinder its effectiveness when combined with MCTS. We will continue to make efforts and try our best to incorporate reliable outcomes with MCTS in the new version. Besides this, we believe that the current results with greedy decoding and sampling decoding already convincingly underscore the superiority of T2TCO.
>
> **Q13: Do you plan to release the source code during the rebuttal period?**
>
> Thanks. We will release the source code upon the acceptance of the paper.
>
> ---
> We hope this response could help address your concern, and wish to receive your further feedback soon.

---

> > ### Comment · Reviewer_gU16 · 2023-08-11
> >
> > Thanks for your rebuttal. Most of the concerns are well addressed. However, I'm still curious about the generalization performance of the diffusion-based methods (e.g., DIFUSCO and T2TCO) on the distribution (e.g., cluster, gaussian, etc., instead of uniform) setting. Moreover, it would be better to include results on large-scale benchmark instances since you already consider large-scale synthetic ones in the main experiments.

---

> > > ### Author Response · Authors · 2023-08-12
> > > **Supplementary results on distribution generalization and large-scale benchmark instances.**
> > >
> > > Thanks for the prompt reply and constructive advice. In this response, we supplement new results on distribution generalization and large-scale TSPLIB instances to further validate the effectiveness of T2TCO.
> > >
> > > (1) The generalization results of the diffusion-based methods (e.g., DIFUSCO and T2TCO) on the distribution (e.g., cluster, gaussian, etc., instead of uniform) are presented below. The TSP data of different distributions are from [1].
> > >
> > > Generalization results from the model trained with TSP100-Uniform data:
> > >
> > > | Methods | Decoding | TSP100-Uniform | TSP100-Gaussian |
> > > | ------- | -------- | -------------- | --------------- |
> > > | DIFUSCO | Greedy   | 7.78, 0.20%    | 5.72, 0.72%     |
> > > | T2TCO   | Greedy   | 7.76, 0.06%    | 5.69, 0.28%     |
> > > | DIFUSCO | Sampling | 7.76, 0.00%    | 5.68, 0.09%     |
> > > | T2TCO   | Sampling | 7.76, 0.00%    | 5.68, 0.02%     |
> > >
> > > Generalization results from the model trained with TSP500-Uniform data:
> > >
> > > | Methods | Decoding | TSP500-Uniform | TSP500-Rotation | TSP500-Explosion | TSP300-Cluster_3_10 | TSP300-Cluster_7_50 |
> > > | ------- | -------- | -------------- | --------------- | ---------------- | ------------------- | ------------------- |
> > > | DIFUSCO | Greedy   | 16.82, 1.63%   | 12.69, 2.44%    | 12.06, 2.80%     | 9.72, 2.64%         | 5.80, 3.09%         |
> > > | T2TCO   | Greedy   | 16.68, 0.78%   | 12.51, 1.00%    | 11.94, 1.76%     | 9.61, 1.49%         | 5.75, 2.17%         |
> > > | DIFUSCO | Sampling | 16.68, 0.81%   | 12.60, 1.65%    | 11.94, 1.78%     | 9.62, 1.57%         | 5.75, 2.18%         |
> > > | T2TCO   | Sampling | 16,62, 0.42%   | 12.45, 0.48%    | 11.87, 1.18%     | 9.55, 0.85%         | 5.72, 1.67%         |
> > >
> > >
> > > [1] Towards Omni-generalizable Neural Methods for Vehicle Routing Problems. In ICML 2023.
> > >
> > > (2) We supplement the results (optimality drop) on large-scale TSPLIB benchmark instances with 200-1000 nodes. For each instance, we normalize the coordinates to [0,1] and solve with the solving algorithms.
> > >
> > > | Instances | DIFUSCO | T2TCO |
> > > | --------- | ------- | ----- |
> > > | pr439     | 1.28%   | 0.26% |
> > > | pcb442    | 0.00%   | 0.00% |
> > > | fl417     | 0.00%   | 0.00% |
> > > | tsp225    | 0.99%   | 0.59% |
> > > | u724      | 0.00%   | 0.00% |
> > > | pr299     | 0.57%   | 0.14% |
> > > | rd400     | 0.12%   | 0.00% |
> > > | rat575    | 0.83%   | 0.26% |
> > > | a280      | 1.11%   | 0.00% |
> > > | pr264     | 1.12%   | 0.48% |
> > > | d493      | 0.00%   | 0.00% |
> > > | p654      | 0.00%   | 0.00% |
> > > | d657      | 0.00%   | 0.00% |
> > > | rat783    | 1.67%   | 1.22% |
> > > | ts225     | 0.32%   | 0.32% |
> > > | lin318    | 0.30%   | 0.30% |
> > > | u574      | 0.00%   | 0.00% |
> > > | pr1002    | 2.21%   | 1.42% |
> > > | pr226     | 0.34%   | 0.22% |
> > > | **Mean**      | **0.57%**   | **0.27%** |
> > >
> > > We hope this response could help address your remaining concern, and wish to receive your further feedback soon.

---

> > > > ### Comment · Reviewer_gU16 · 2023-08-12
> > > > **No further issues**
> > > >
> > > > Thanks for the additional experiments. The results are pretty impressive. I raise my score to 6.

---

> > > > > ### Author Response · Authors · 2023-08-15
> > > > > **Thanks for increasing your rating.**
> > > > >
> > > > > Thank you very much for increasing your rating to 6, and we are glad to know you appreciate our response with impressive new results. Your valuable insights have played a pivotal role in enhancing the comprehensiveness of our paper.
> > > > >
> > > > > Also, if possible, we would like to know if there is any concern left that makes it hard for you to further increase the rating. We are more than happy to further address any potential concerns.

---

> > > > > > ### Comment · Reviewer_gU16 · 2023-08-18
> > > > > >
> > > > > > Dear authors: I don't have any further concerns. The evaluation is based on the potential impact of this paper, which is indeed a subjective criterion. But no worries, this paper has already got quite positive scores.

---

> > > > > > > ### Author Response · Authors · 2023-08-18
> > > > > > > **Thanks again**
> > > > > > >
> > > > > > > We sincerely appreciate your valuable engagement and the time you dedicated to reviewing our work. Thank you once more!

---

### Official Review · Reviewer_uaDN · 2023-06-27

**Soundness:** 3 good
**Presentation:** 3 good
**Contribution:** 3 good
**Rating:** 6
**Confidence:** 4

**Summary:**

This paper proposed a diffusion model based solution distribution learning method for combinatorial optimization. The main novelty is that a search mechanism is proposed, which uses the learned distribution to conduct a gradient based search for each instance where the gradient is computed w.r.t the optimiztion objective function. Comparing with existing testing-time gradient update methods such as active search and meta-learning, this method does not need to update model parameters, hence could be more efficient. Experiments on two graph problems TSP (up to 1000 nodes) and MIS show that the proposed method outperforms recent neural CO methods.

**Strengths:**

1. The idea of using learned distribution to perform search is interesting and novel as far as I know.

2. Empirical performance on two important problems TSP and MIS is impressive.

**Weaknesses:**

1. While the paper is interesting, a weakness is that the authors did not clearly state what assumptions they made, or the proposed approach is suitable in solving which kinds of problems. For example:

a. The decision variables are binary. Though it is possible to transform non-binary problems into binary ones, a large number of variables could be involved, making the proposed approach impractical.

b. The constraints can be transformed to some sort of penalty with meaningful gradient.

c. A problem-specific post-processing procedure is needed to transform the prediction into feasible solutions.

The authors only evaluate the proposed method on two simple problems TSP and MIS, which satisfies the above conditions. But it is important to demonstrate its effectiveness on more challenging and practical problems.

2. The distribution learning method in Section 3.2 heavily relies on [19]. The proof of Proposition 1 is adapted from [39]. The authors should be precise about what are the novel techniques and which parts are borrowed from existing works.

3. Some design and implementation details lacks explanation. Please see the below questions.




**Questions:**

1. What kind of problems can the proposed approach solve? Specifically, what kind of variables and constraints, and is it easy to design a problem-specific post-processing procedure?

2. How is the diffusion model in Section 3.2 different from the one in Difusco [19]?

3. Some questions about design and implementation:

a. For TSP, why and how is 2OPT applied? It looks like 2OPT is independent from the proposed gradient search mechanism. In this case, it is a general solution improvement technique and should be applied to other baselines for fair comparison. Moreover, better solution quality with 2OPT is not suprising and does not help in justifying the contribution of the proposed method.

b. The authors state that "For gradient search, T2TCO generally involves 3 iterations with 10 guided denoising steps for each". Why you need 3 iterations and how about the performance of only 1 iteration? How do you perform the 3 iterations, independntly and then picking the best solution?

c. The authors used k=50 for graph sparsification. How sensitive does the proposed method w.r.t. to k?

**Limitations:**

As mentiond above, this paper lacks a discussion on the applicability of the proposed approach, i.e. it is suitable for what kind of problems. The authors also did not explicitly describe the limitations of the proposed method.

---

> ### Author Rebuttal · Authors · 2023-08-09
>
> Thanks for the valuable comments, constructive advice, and recognition of our idea and empirical results. Below we respond to your specific comments.
>
> **Q1: What kind of problems can the proposed approach solve? Specifically, what kind of variables and constraints, and is it easy to design a problem-specific post-processing procedure?**
>
> We sincerely appreciate your valuable summary concerning the types of problems that T2TCO can effectively manage. We would like to reorganize it and provide explanations as follows:
>
> Mandatory Requirements:
> 1. The decision variables are limited to a countable and finite set of values. Currently, we focus on binary decision variables and the model predicts a $N\times 2$ heatmap for the solution. But it can be easily extended to $N\times k$ prediction through the multinomial diffusion. Note that even binary variables can cover a wide range of CO problems especially on graphs, since most problems can be attributed to edge-selecting problems like TSP, CVRP, and node-selecting problems like MIS, MAX Clique.
> 2. A problem-specific post-processing procedure is required to transform the prediction into feasible solutions. For problems featuring binary decision variables, this prerequisite can be easily met with the greedy decoding strategy, which sequentially inserts variables (edges or nodes) with the highest confidence if there are no conflicts.
>
> Nice-to-have Requirements:
> 1. The constraints can be transformed to some sort of penalty with meaningful gradient. While it is preferable to include a constraint penalty term in Eq. 7, the utilization of the optimization objective alone is also effective (as evidenced by the TSP results). This is due to the diffusion's capacity to implicitly learn constraints.
>
>
> **Q2: The authors should be precise about what are the novel techniques and which parts are borrowed from existing works.**
>
> Our method is technically based on the diffusion model as the implementation foundation inspired by DIFUSCO [1]. Thus it is unavoidable to follow the diffusion modeling design in the methodology with DIFUSCO [1], whose main idea is to learn the solution distribution given an input instance using diffusion. However, the proposed framework including the test-stage gradient-based search with instance-wise objective feedback and the sequential local improvement mechanism obviously extends beyond DIFUSCO, as also evidenced by the empirical results.
>
> Indeed, we have already marked in the corresponding section when we adopt conclusions or modeling paradigms from previous works. The diffusion modeling paradigm in Sec. 3.2 follows DIFUSCO [1] as marked in Line 149, while we differ DIFUSCO by explicitly modeling the solving task as a conditional generation task. Thus the derivation in Sec. 3.2 is based on the conditional context of diffusion models. Prop. 1 is adapted from [39] as marked in Line 205, which we utilize as an existing conclusion to incorporate explicit objective optimization in generation.
>
> We will enhance the clarity of our references in the new version of our work.
>
>
> **Q3: For TSP, why and how is 2OPT applied? 2OPT should be applied to other baselines for a fair comparison.**
>
> 2OPT is indeed independent from the proposed gradient search mechanism as a general solution improvement technique. We have supplemented the experiments that apply the 2OPT solution improvement technique to other main baselines in original Table 1 and 2 for fairer comparison. Please refer to the general response.
>
> **Q4: Why do you need 3 iterations and how about the performance of only 1 iteration? How do you perform the 3 iterations?**
>
> On one hand, each iteration contributes to a single process of solution improvement, and employing more iterations can lead to improved solutions. On the other hand, too many iterations could lead to additional computational overhead. As a result, we aim to strike a balance in this trade-off and have opted to set the iteration number to 3.
>
> We supplement TSP-500 Greedy results of the hyperparameter #iterations below.
>
> | #iterations | Tour Length $\downarrow$ | Drop $\downarrow$ |
> | ------ | ------- | ---- |
> | 0  | 16.813      | 1.59% |
> | 1  | 16.705      | 0.94% |
> | 3  | 16.683      | 0.80% |
> | 5  | 16.678      | 0.77% |
> | 10 | 16.669      |  0.72% |
>
>
> The iterations are performed sequentially, where each iteration is based on the previous iteration's result to improve. As claimed in Line 230-232, each iteration involves adding a certain degree of noise to disrupt the structure, denoising with objective gradient guidance to obtain a lower-cost soft-constrained solution, and subsequently post-processing to decode a feasible solution.
>
>
> **Q5: How sensitive does the proposed method w.r.t. to graph sparsification hyperparameter $k$?**
>
> We supplement TSP-500 Greedy inference results of graph sparsification hyperparameter $k$ below.
>
> | $k$ | Tour Length $\downarrow$ | Drop $\downarrow$ |
> | -- | --- | --|
> | 10  | 16.687      | 0.83% |
> | 50  | 16.681      | 0.79% |
> | 100  | 16.678      | 0.77% |
> | 200  | 16.674      | 0.74% |
>
> A larger value of $k$ can yield improved results, yet it can also lead to increased computational overhead. Thus, we strike a balance in this trade-off and set $k$ to 50.
>
> ---
> We hope this response could help address your concern, and wish to receive your further feedback soon.
>
> ---
>
> **Reference:**
>
> [1] DIFUSCO: Graph-based Diffusion Solvers for Combinatorial Optimization. arXiv 2023.

---

> > ### Comment · Reviewer_uaDN · 2023-08-17
> >
> > Thanks for the response. I urge the authors to incorporate the response content in the paper. I have increased my score to 6.

---

> > > ### Author Response · Authors · 2023-08-18
> > > **Thanks for increasing your rating.**
> > >
> > > We will continue to polish the paper and integrate the response content to elevate the comprehensiveness of our paper. Thanks once more for your time and effort in reviewing our paper!

---

### Official Review · Reviewer_p83a · 2023-07-06

**Soundness:** 3 good
**Presentation:** 3 good
**Contribution:** 3 good
**Rating:** 7
**Confidence:** 4

**Summary:**

The paper proposes a training to testing (T2TCO) framework to better align with the goal of combinatorial optimization (CO) problems, instead of pursuing the average expected performance, Real-world CO problems care more about instance-wise optimal solution. And thus, the paper first models the solution distribution with generative diffusion model and further conducts gradient search during testing to fine-tune the results. The framework design is novel, interesting, and sound. Experiments show the significant superiority of the framework.

**Strengths:**

The paper is well written. The motivation of the paper aims towards the goal of per-instance optimality for CO problems and introduces a novel T2TCO framework to resolve the problems. The gradient-search on each testing instance requires no longer fine-tuning the parameters of the model, which is also an interesting improvement.
The paper conducts experiments on two representative CO problems and both achieves better results than the state-of-the-art methods.


**Weaknesses:**

The only concern I could think about is the cross-domain generalization ability or even the in-distribution generalization ability of the supervised learning framework for CO problems.

**Questions:**

1.	Is there any comparison on the training resource requirement / training time between T2TCO and other state-of-the-art methods listed in the paper? Could the authors provide some experiment data on that to give the readers a feeling about the training difficulty of the framework? (e.g. How many GPU/CPU hours, how much space, how many data does T2TCO require to converge, how about the other methods listed in the paper?)

**Limitations:**

The motivation is practical, the framework is novel, interesting and the methodology is sound. Therefore, I omit the limitation of this paper.

---

> ### Author Rebuttal · Authors · 2023-08-09
>
> Thanks for the valuable comments and for acknowledging the motivation practicality, framework novelty, and methodology soundness of our paper. Below we respond to your specific comments. Since your concerns are representative and common, we have discussed them mainly in the general response.
>
> **Q1: Cross-domain generalization ability evaluation.**
>
> Thanks for the comment. We have supplemented the generalization results in the general response.
>
> **Q2: More details about the training difficulty of the framework, e.g., the training resource requirement, training time, and data requirement.**
>
> We have provided the details about the offline training resource requirement of T2TCO framework in the general response.
>
>
> ---
> We hope this response could help ease your concern and wish to receive your further feedback soon.

---

> > ### Comment · Reviewer_p83a · 2023-08-14
> > **Reply to the authors**
> >
> > Dear authors, thanks for your time in the rebuttal. My questions have been well-addressed, and thus I would keep my score.

---

> > > ### Author Response · Authors · 2023-08-18
> > > **Thanks**
> > >
> > > Glad to know your concerns have been addressed, and thank you for your time and effort in reviewing our work.

---

### Official Review · Reviewer_uUyD · 2023-07-09

**Soundness:** 3 good
**Presentation:** 2 fair
**Contribution:** 3 good
**Rating:** 6
**Confidence:** 4

**Summary:**

The paper presents T2TCO, a new approach for neural combinatorial optimization based on using gradient-based search in the solution  space during test time. The proposed approach leverages generative modeling (specifically diffusion) for graduated solution improvement but employs an additional energy-based model (EBM) to evaluate the compatibility of solutions with the optimal solution. Experiments show improvement over DIFUSCO (similar generative modeling approach but without test-time gradient-based search) and a range of baselines in two popular CO benchmarks: TSP and MIS.

**Strengths:**

- Interesting work suggesting a new approach for ML4CO. The approach extends DIFUSCO (or similar approaches) with gradient-based search during test time to improve performance.
- Experiments show the proposed approach outperforms the baselines. In particularly, it outperforms the recent DIFUSCO which it extends with gradient-based test-time search.

**Weaknesses:**

- Related literature: there are several relevant works that are not being mentioned and discussed. First, gradient-based search in prediction/inference (test-time) for different purposes has been proposed before in a range of settings. e.g., in [1], [2], [3]. While using it in combinatorial optimization is new (to my knowledge), the principles have been proposed before and related previous works should be mentioned and similarities and differences highlighted. Second, using diversity-focused solution generation (test-time) in combinatorial problems has been studied before e.g., in [4] and [5]. Third, introducing constraints to gradient-based search by continuous relaxations has been done before, e.g., in [6] and [7].
	* related to the above, it is important to mark what parts of the work is based on previous work. In particular, it sounds like Section 3.2 is primarily based on DIFUSCO, which if indeed is the case, should be clearly stated.

- Experiments: (1) No experiments on generalization: it sounds like each group of instances (e.g., TSP-100) was trained and tested on instances from the same size. Previous work on ML4CO has often looked at the ability of networks trained on smaller sizes to generalize well for larger problems (including in DIFUSCO [8]). It is not clear how the proposed gradient-based search approach works in such settings. (2) While experiments show consistent improvement over previous SOTA (usually DIFUSCO), the improvement is often somewhat modest (except for MIS with ER graphs) and there is no clear analysis to demonstrate where the gain are comping from (e.g., is it consistent but modest gain across all instances vs. a small number of instances that got significant boost).

- The proposed method requires manual relaxation of constraints to penalty terms and therefore is not clearly generic to all CO problems (and it is not clear that it will work well across relaxations). This is not a critical issue but it would be interesting to cover a set of popular types of constraints and their relaxations and make sure they are tested. Potentially, this can be based on previous literature (e.g., [6] where a list of continuous relaxations for popular hard constraints is provided).

- Not a critical issue, but the paper seems to build extensively on DIFUSCO that, as far as I can see, is not yet been published in a peer-reviewed venue.

Minor:
- line 51: desired to estimate -> "designed to estimate"?
- line 299: I think "TSP-50" here should be "TSP-500"
- line 310: "T2TCO is not that sensitive to the quality of labels in training" - I think this statement is too strong given that the quality of labels seems to have significant impact on the performance (it may be that it is less sensitive than other approaches but it is still sensitive).


[1] Belanger, D., Yang, B., & McCallum, A. (2017, July). End-to-end learning for structured prediction energy networks. In International Conference on Machine Learning (pp. 429-439). PMLR.

[2] Chen, Y., Shi, Y., & Zhang, B. (2019, February). Optimal Control Via Neural Networks: A Convex Approach. In International Conference on Learning Representations.

[3] Amos, B., Xu, L., & Kolter, J. Z. (2017, July). Input convex neural networks. In International Conference on Machine Learning (pp. 146-155). PMLR.

[4] Cohen, E., & Beck, J. C. (2021). Heavy-Tails and Randomized Restarting Beam Search in Goal-Oriented Neural Sequence Decoding. In Integration of Constraint Programming, Artificial Intelligence, and Operations Research: 18th International Conference, CPAIOR 2021, Vienna, Austria, July 5–8, 2021, Proceedings 18 (pp. 115-132). Springer International Publishing.

[5] Bai, Y., Zhao, W., & Gomes, C. P. (2021). Zero Training Overhead Portfolios for Learning to Solve Combinatorial Problems. arXiv preprint arXiv:2102.03002.

[6] Chen, Di, et al. "Deep reasoning networks for unsupervised pattern de-mixing with constraint reasoning." International Conference on Machine Learning. PMLR, 2020.

[7] Chen, Di, et al. "Automating crystal-structure phase mapping by combining deep learning with constraint reasoning." Nature Machine Intelligence 3.9 (2021): 812-822.

[8] Sun, Z., & Yang, Y. (2023). Difusco: Graph-based diffusion solvers for combinatorial optimization. arXiv preprint arXiv:2302.08224.


**Questions:**

I would appreciate the authors' response on my main concerns listed above.

**Limitations:**

While there is no explicit discussion of limitation, I have no concerns regarding the impact of the manuscript.

---

> ### Author Rebuttal · Authors · 2023-08-09
>
> Thanks for the thorough review and valuable comments. We are encouraged with your acknowledgement to our idea, experiments and contribution to the ML4CO community. Below we respond to your specific comments.
>
> **Q1: Some related works are missed.**
>
> Thanks for sharing these related works. We will disscuss these works in our new version.
>
> We would like to note that the mentioned principles can be somehow general under various settings. For instance, a broad spectrum of diffusion models and energy-based models fall within the overarching principle of employing gradient-based sampling during the inference phase. While these approaches share a common design feature, they can technically exhibit significant variations from one another, and the uniqueness and novelty inherent in each of these methods could still be appreciated.
>
> When compared to the mentioned related works, including SPENs [1], ICNNs [2], or even general score-based (diffusion) models, concerning the gradient utilization in the test phase, T2TCO similarly train a network to estimate the score funtion $\nabla_x \log(q(x_t))$, which can be seen as the gradient of some sort of energy function. Subsequently, the sampling in the test phase utilizes the estimated scores to sample a valid generation. However, T2TCO goes a step further by incorporating an explicit objective optimization term during the test phase. The guidance of the trained network and the guidance from the objective gradient are organically integrated to enable optimization in the solution space. When it comes to specific framework designs, T2TCO is obviously different from the above methods with the sequential local improvement machanism.
>
> **Q2: It sounds like Section 3.2 is primarily based on DIFUSCO, which if indeed is the case, should be clearly stated.**
>
> Actually, we have already marked in the corresponding section when we adopt conclusions or modeling paradigms from previous works. The diffusion modeling paradigm in Sec. 3.2 follows DIFUSCO [3] as marked in Line 149, while we differ DIFUSCO by explicitly modeling the solving task as a conditional generation task. Thus the derivation in Sec. 3.2 is based on the conditional context of diffusion models. We will enhance the clarity of our references in the new version of our work.
>
>
> **Q3: No experiments on generalization.**
>
> Thanks for the comment. We have supplemented the generalization results in the general response.
>
> **Q4: While experiments show consistent improvement over previous SOTA, the improvement is often somewhat modest. Where the gains are coming from?**
>
> Firstly, we would like to emphasize that the performance gain achieved by T2TCO over the previous methods is actually significant among the SOTAs, rather than minor. In the TSP experiments, we observed an average performance gain of 49.3% across various settings compared to the previous state-of-the-art. Although the absolute numerical improvement might not appear exceedingly large, it is important to consider that the previous SOTA methods had already achieved high quality results (with less than 1% optimality drop). Please kindly note that reviewers p83a, uaDN, and gU16 all acknowledge the empirical results of T2TCO.
>
> For the performance gains on individual instances, please see Table 5 in the supplemented pdf for a direct perception, where we list the model performance on each instance of TSPLIB dataset. Since DIFUSCO have reached or approached optimality in partial instances, the performance gains are distributed among other instances with certain variation.
>
>
> **Q5: Manual relaxation of constraints to penalty terms is not clearly generic to all CO problems. This is not a critical issue but it would be interesting to cover a set of popular types of constraints and their relaxations and make sure they are tested.**
>
> Thanks for providing the related work and offering the advice. Actually, T2TCO is designed to learn the constraints via latent space learning (i.e. generative modeling) which allows the loss for gradient computing in test time. It only involves the raw objective function as we did for the TSP case (see Line 202-203). We optionally add the constraint penalty term for the MIS case, which is designed mainly as an auxiliary support to latent space learning for constraints.
>
> **Q6: Minor issues and suggestions.**
>
> We sincerely appreciate your thorough review. We will revise the manuscript accordingly to elevate it into a more refined paper.
>
> ---
> We hope this response could help address your concerns, and wish to receive your further feedback soon.
>
> ---
>
> **Reference:**
>
> [1] End-to-end learning for structured prediction energy networks. ICML 2017.
>
> [2] Input convex neural networks. ICML 2017.
>
> [3] DIFUSCO: Graph-based Diffusion Solvers for Combinatorial Optimization. arXiv 2023.

---

> > ### Comment · Reviewer_uUyD · 2023-08-17
> > **Author response**
> >
> > Thank your for your response and additional experimental results. I have increased my score.

---

> > > ### Author Response · Authors · 2023-08-18
> > > **Thanks for increasing your rating.**
> > >
> > > Thank you very much for increasing your rating to 6, and for acknowledging our response and additional experimental results. We sincerely appreciate your time and effort in reviewing our work!

---

### Author Rebuttal · Authors · 2023-08-09

Dear Area Chairs and Reviewers,

We greatly appreciate the reviewers' time, valuable comments, and constructive suggestions. We are delighted that all the reviews have expressed a positive inclination towards accepting our submission. Overall, the reviewers deem our paper as "well written" (p83a, gU16) and "constructive for ML4CO community" (uUyD, gU16), acknowledging our methodology as "interesting" (uUyD, p83a, uaDN), "novel" (p83a, uaDN, gU16), and "sound" (p83a, gU16), with "clear" and "practical" motivation (p83a, gU16) and "significant" or "impressive" empirical results (p83a, uaDN, gU16).

In the author response period, we make every effort to address reviewers’ concerns and provide additional experimental results in the **Supplemented PDF** to further verify our contributions. The summary of our main experimental efforts corresponding to the supplemented pdf is presented as follows:

(1) We provide experimental results on generalization. Specifically, based on the problem set {TSP-50, TSP-100, TSP-500, TSP-1000}, we train the model on a specific problem scale and then evaluate it on all problem scales. Table 1 and 2 present the generalization results of T2TCO and DIFUSCO [1] with greedy decoding and sampling decoding, respectively. The results show the satisfying cross-domain generalization ability of T2TCO, e.g., the model trained on TSP-1000 achieves less than 0.8% optimal gap on all other problem scales.

(2) We apply the 2OPT solution improvement technique to other main baselines in the original Table 1 and Table 2 for a fairer comparison. Table 3 presents the results on TSP 500 and 1000, and Table 4 presents the results on TSP 50 and 100. In Table 3, we mainly include DIMES and DIFUSCO for comparison, as they are the only two competitive methods in the non-2OPT setting. The results reaffirm T2TCO's superiority, just as in previous findings.

Remark: T2TCO and DIFUSCO's results on TSP-50 have been updated in Table 4, since we observe that the previous TSP-50 results were obtained from the models trained on TSP-100. We sincerely apologize for this oversight, and we have verified other results to ensure their accuracy.

(3) We evaluate our model trained with random 100-node problems on real-world TSPLIB instances with 50-200 nodes. The compared baselines include DIFUSCO and baselines listed in [2]'s Table 3. Table 5 presents the results and shows T2TCO's superiority achieving merely 0.133% optimality gap, 58.3% better than the best previous method.

(4) We provide the details about the offline training resource requirement of T2TCO framework in Table 6. The results are calculated on A100 GPUs. For comparison reference: AM [3] requires 128M instances generated on the fly for training TSP-100, and it requires 45.8h on 2 1080Ti GPUs; POMO [4] requires 200M instances generated on the fly for training TSP-100, and it requires ~ 1 week on a single Titan RTX; Sym-NCO [5] based on POMO requires ~ 2 weeks on a single A100, Sym-NCO [5] based on AM requires ~ 3 days on 4 A100 GPUs.


In our individual responses, we provide detailed answers to all the specific questions raised by the reviewers. We hope these responses could help address the reviewers' concerns, and further discussions are welcomed towards a comprehensive evaluation of our work.

---

**Reference:**

[1] DIFUSCO: Graph-based Diffusion Solvers for Combinatorial Optimization. arXiv 2023.

[2] Graph Neural Network Guided Local Search for the Traveling Salesperson Problem. ICLR 2022.

[3] Attention, learn to solve routing problems. ICLR 2019.

[4] POMO: Policy Optimization with Multiple Optima for Reinforcement Learning. NeurIPS 2020.

[5] Sym-NCO: Leveraging Symmetricity for Neural Combinatorial Optimization. NeurIPS 2022.

---

### Decision · Program_Chairs · 2023-09-21

**Decision:**

Accept (poster)

**Comment:**

The paper proposes a training to testing framework, which leverages the generative model to learn solution distribution and then conducts a gradient-based search within the solution space during testing. All the reviewers agreed that the problem is well-motivated, the proposed approach is novel, and the empirical evaluations are thorough. Overall, this paper is well written and easy to follow, and the idea of training to testing is valuable for this community. The authors' rebuttal addressed most of the concerns of reviewers, and all the reviews tend to accept the paper during the reviewer-author discussion phase. Therefore, I recommend this paper to the NeurIPS 2023 conference.